# Revisiting PCA for Time Series Reduction in Temporal Dimension

## Abstract

Deep learning has significantly advanced time series analysis (TSA), enabling the extraction of complex patterns for tasks like classification, forecasting, and regression. While dimensionality reduction has traditionally focused on the variable space—achieving notable success in minimizing data redundancy and computational complexity—less attention has been paid to reducing the temporal dimension. In this study, we revisit Principal Component Analysis (PCA), a classical dimensionality reduction technique, to explore its utility in temporal dimension reduction for time series data. It is generally thought that applying PCA to the temporal dimension would disrupt temporal dependencies, leading to limited exploration in this area. However, our theoretical analysis and extensive experiments demonstrate that applying PCA to sliding series windows not only maintains model performance but also enhances computational efficiency. In auto-regressive forecasting, the temporal structure is partially preserved through windowing, and PCA is applied within these windows to denoise the time series while retaining their statistical information. By preprocessing time series data with PCA, we reduce the temporal dimensionality before feeding it into TSA models such as Linear, Transformer, CNN, and RNN architectures. This approach accelerates training and inference and reduces resource consumption. Notably, PCA improves Informer training and inference speed by up to 40% and decreases GPU memory usage of TimesNet by 30%, without sacrificing model accuracy. Comparative analysis against other reduction methods further highlights the effectiveness of PCA in enhancing the efficiency of TSA models. Code is provided in the supplementary materials.

## 1 Introduction

Time series analysis (TSA) plays a pivotal role across various fields (Van Zyl et al., 2024; Hittawe et al., 2024), owing to its ability to extract valuable information from sequential data, facilitating accurate predictions and classifications. Recent advancements in the field have witnessed the emergence of sophisticated deep-learning models (Eldele et al., 2024; Wu et al., 2023; Zhou et al., 2021) designed to effectively analyze time series data.

Dimensionality reduction techniques have been successfully applied to reduce complexity in time series data, but their focus has primarily been on the variable dimension (Xu et al., 2023; Hyndman et al., 2015). These methods, which aim to minimize redundancy in variable space, have been effective in reducing computational complexity and improving model performance. However, far less attention has been given to reducing the temporal dimension, despite the potential benefits of alleviating the burdens associated with processing long time series.

The time series lengths are generally larger than the number of the variable sizes, suggesting that temporal dimensionality reduction should provide better compression. The long time series is segmented to time series windows in the auto-regressive forecasting task, and recent studies show that larger window length includes more temporal information and hence brings better forecasting results (Zeng et al., 2023; Nie et al., 2022). Therefore, a fundamental paradox arises: the contradiction between the length of the series windows and the ease of TSA model learning. While longer windows provide more information, they also increase the difficulty of model learning: raw series data inherently contains redundancy (Li et al., 2023; Prichard & Theiler, 1995), and inputting such data can significantly increase both the computational and spatial burdens associated with model

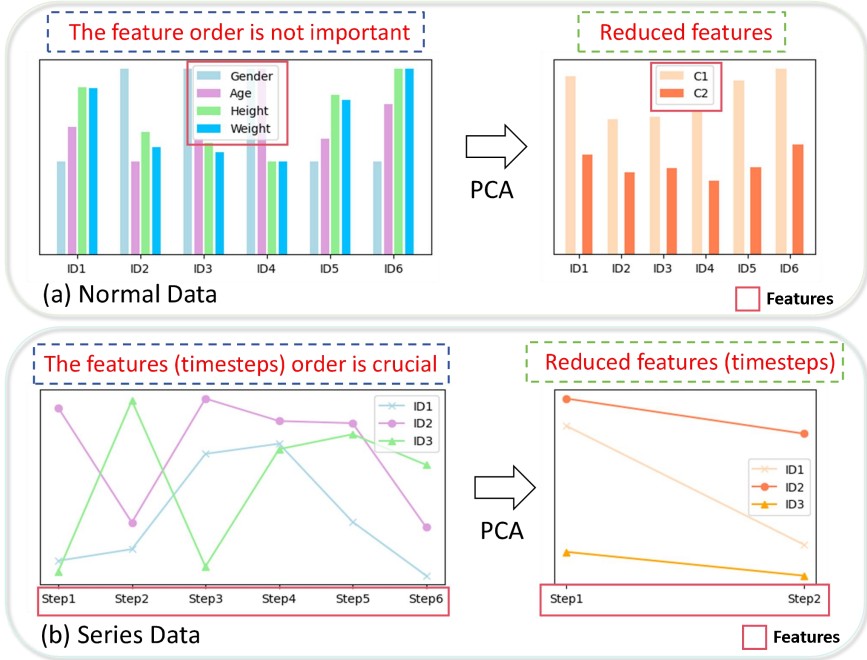

Figure 1: When PCA is applied to normal data, the order of data features is irrelevant, and there is no temporal correlation between features, as shown in (a); Our research demonstrates that PCA can also be applied to time series data, where the order of features (time steps) is significant and there is temporal correlation, as illustrated in (b).

training and inference. This challenge persists across TSA models, from RNNs (Hewamalage et al., 2021) and CNNs (Wu et al., 2023) to Transformers (Wen et al., 2022).

To the best of our knowledge, there has been no systematic method for compressing time series data in temporal dimension while preserving critical information before feeding it into deep-learning models. Simple downsampling the series inevitably results in information loss within time series data, while merely shortening the input window leads to degraded prediction performance (Nie et al., 2022; Gao et al., 2023). Existing time series feature extraction methods (Ye & Keogh, 2009; Schäfer & Leser, 2017) are typically tailored to specific tasks like classification and are computationally expensive, prioritizing performance over efficiency.

To address these challenges, we explore the use of Principal Component Analysis (PCA) (Pearson, 1901) for feature extraction and temporal dimensionality reduction of time series data before inputting it into deep-learning models. PCA is effective at capturing essential information while reducing data dimensionality across various domains (Gewers et al., 2021). The PCA algorithm identifies principal components that represent directions of maximal variance, and projecting the data onto these components extracts fundamental features and uncovers latent patterns. However, PCA's application to time series preprocessing has been underutilized due to the unique characteristics of time series data. In normal data, the order of features is irrelevant, and there is no temporal correlation between features, as shown in Fig. 1 (a). In contrast, in time series data, each time step can be considered a "feature", and the sequential order of these features (time steps) is crucial, introducing temporal correlations, as illustrated in Fig. 1 (b). Although PCA might disrupt the temporal structure, it also refines the information by retaining key statistical characteristics and reducing overfitting and redundancy, which enhances training efficiency without compromising model performance.

We propose that PCA is well-suited for compressing redundant time series in temporal dimension and extracting salient features. Specifically, PCA mitigates noise and redundancy by isolating key features and reducing correlations among different time steps, thereby lowering the risk of overfitting in deep-learning models. Additionally, by analyzing the entire training dataset, PCA captures shared patterns and maps time series onto principal components that represent common features. This process preserves key statistical information while shortening the original series. Therefore, preprocessing time series with PCA before inputting it into deep-learning models can alleviate computational burdens, reduce learning difficulty, while retaining performance in TSA tasks.

To substantiate our hypothesis, experiments are conducted on three typical TSA tasks: time series classification (TSC) (Middlehurst et al., 2024), time series forecasting (TSF) (Chen et al., 2023), and time series extrinsic regression (TSER) (Mohammadi Foumani et al., 2024). Four types of advanced deep-learning based TSA models are evaluated: MLP(linear)-based model (Zeng et al., 2023), Transformer-based models (Zhou et al., 2021; 2022; Nie et al., 2022), CNN-based model (Wu et al., 2023), and RNN-based models (Chung et al., 2014; Hochreiter, 1997). The findings show that PCA preprocessing maintains model performance while reducing training/inference burdens across all assessed TSA models and tasks. Additionally, PCA outperforms other dimensionality reduction methods, such as shortening/downsampling the historical input series and adding a linear/1D-CNN dimensionality reduction layer before the original TSA model.

In summary, the contribution of this study is threefold:

- This study initially establishes the utility of PCA as an efficient tool for time series reduction in temporal dimension in the domain of TSA. We conduct theoretical discussions on PCA's effectiveness in denoising time series and preserving their statistical information, providing a theoretical foundation for its efficacy.
- This study integrates PCA with advanced deep-learning models for TSA, including Linear, Transformer, CNN, and RNN models, demonstrating its ability to reduce computational costs and memory requirements. Notably, PCA accelerates Informer's training and inference by up to 40% and decreases TimesNet's GPU memory usage by 30%. These results highlight the generalization of PCA's application across various TSA models.
- We apply PCA across diverse TSA tasks: classification, forecasting, and extrinsic regression, proving its versatility in different applications.

## 2 RELATED WORK

**Time Series Analysis Models.** Traditional TSA models like ARMA and ARIMA (Box et al., 2015) rely on statistical foundations, assuming linear relationships between past and present observations to discern patterns in the time series data. However, the rise of deep-learning models has gained significant attention in TSA due to their enhanced expressive capability and ability to effectively utilize available data, leading to improved performance over traditional statistical models. RNN-based models (Chung et al., 2014; Hochreiter, 1997) are initially employed in TSA tasks due to their capability to process sequences and capture temporal dependencies. However, due to their inherent difficulties in propagating gradients through many time steps, RNN-based models often encounter issues of gradient vanishing or gradient explosion (Hanin, 2018). CNN-based models (Liu et al., 2022; Luo & Wang, 2024) constitute a major branch within deep-learning, aiming to capture temporal dependencies through convolutional layers. Notably, TimesNet (Wu et al., 2023) transforms 1D time series into a set of 2D tensors based on multiple periods and employs a CNN structure to extract features from these tensors. Transformer-based models have also found widespread application in TSA, leveraging self-attention (Vaswani et al., 2017) to capture long-term dependencies across different time steps. Informer (Zhou et al., 2021) achieves a complexity reduction to $O(L \log L)$ by replacing the conventional self-attention mechanism with KL-divergence-based ProbSparse attention. FEDformer (Zhou et al., 2022) achieves a complexity reduction to $O(L)$ by employing frequency-domain self-attention through the use of Fourier or wavelet transforms and the random selection of frequency bases. Fredformer (Piao et al., 2024) is designed to address frequency bias in TSF by ensuring equal learning across various frequency bands. PatchTST (Nie et al., 2022) partitions the time series into multiple segments, treating each as a token, and employs an attention module to learn the relationships between these tokens. Additionally, DLinear (Zeng et al., 2023) leverages a linear model to attain noteworthy results in TSF tasks, demonstrating the efficacy of linear models in the domain of TSA. SparseTSF (Lin et al., 2024b) is a lightweight Linear-based model that uses Cross-Period Sparse Forecasting to decouple periodicity and trend, achieving superior performance with fewer parameters.

**PCA applications in various domains.** PCA (Pearson, 1901) has diverse applications in various domains (Marukatat, 2023). In computer vision, (Zhang et al., 2023) proposes a texture-defect detection method using PCA, requiring only a few unlabelled samples and outperforming traditional and deep-learning methods for small and low-contrast defects. (Lin et al., 2024a) introduces a tensor robust kernel PCA model to effectively capture the intrinsic low-rank structure of image data. For

natural language processing, Rémi and Ronan simplify word embeddings via PCA, outperforming existing methods on named entity recognition and movie review tasks (Lebret & Collobert, 2013). In bioinformatics, (Elhaik, 2022) examines the application of PCA in population genetic studies. Moreover, in the domain of engineering, (Hasnen et al., 2023) proposes a PCA-based drift correction method for Nitric Oxides emissions prediction in industrial water-tube boilers.

While PCA has seen extensive use across domains, its application specifically to TSA in temporal dimension has been underexplored until recently. A recent study (Xu et al., 2023) incorporates PCA preprocessing into a Transformer-based forecasting framework to reduce redundant information. However, their approach has several limitations. Firstly, it applies PCA to reduce the variable dimension rather than the temporal dimension, making it similar to conventional PCA applications and not an actual series reduction. Secondly, it is designed for scenarios where a multivariate series forecasts a univariate series, focusing on reducing the variable dimension of covariate series without preprocessing the target variable series, even if the covariate series may have minimal association with the target series. Lastly, its exclusive use of the Transformer model and its focus on forecasting tasks limit the method's applicability to other types of time series models or tasks, thereby restricting its utility for time series data. Other related works involving PCA in TSA (Hyndman et al., 2015; Rea & Rea, 2016) either focus on reducing the variable dimension or on reducing the dimensions of manually extracted features from the original time series, neither of which constitutes actual temporal dimension reduction.

## 3 METHODOLOGY

Given a historical series window $\mathbf{H} = \{X_1, ..., X_L\}$, where $L$ represents the length of the series window, we consider three TSA core tasks. 1) Time Series Classification (TSC): The objective is to predict a discrete class label $\mathbf{C}$ for the series $\mathbf{H}$; 2) Time Series Forecasting (TSF): The goal is to forecast future values of the same series, denoted as $\mathbf{F} = \{X_{L+1}, ..., X_{L+T}\}$, where $T$ is the number of future time steps to predict; 3) Time Series Extrinsic Regression (TSER): In some applications like predicting heart rate from photoplethysmogram and accelerometer data (Tan et al., 2021), neither forecasting nor classification is applicable. TSER involves predicting a single continuous target value $\mathbf{V}$ external to the series based on the input historical series $\mathbf{H}$.

By encapsulating all inputs (historical series window $\mathbf{H}$) and outputs (class label $\mathbf{C}$, future series $\mathbf{F}$ or external target value $\mathbf{V}$), a complete series dataset $D$ can be formed, where $D = [\mathbf{X}; \mathbf{Y}]$. Here, $\mathbf{X}$ is composed of all the historical series, $\mathbf{X} = [\mathbf{H}_1; \ldots; \mathbf{H}_m]$, with $m$ representing the number of samples, and $\mathbf{Y}$ consists of the corresponding targets to be predicted ($\mathbf{C}$, $\mathbf{F}$, or $\mathbf{V}$). The entire dataset $D$ can be split into the training set $D_{\text{train}}$, validation set $D_{\text{val}}$, and test set $D_{\text{test}}$. PCA-related parameters (covariance matrix, eigenvalues, and eigenvectors) are obtained from the training set $D_{\text{train}}$ and subsequently applied to both the validation set $D_{\text{val}}$ and the test set $D_{\text{test}}$ without re-estimation.

### 3.1 ENHANCING TIME SERIES ANALYSIS WITH PCA

Principal Component Analysis (PCA) (Pearson, 1901) is a classical technique for dimensionality reduction and feature extraction. For the training set of the series dataset $D_{\text{train}}$, which consists of $n$ training samples, each with $L$ features (where $L$ is also the length of the series window, and each time step corresponding to a feature), PCA aims to transform the series data into a new coordinate system where the data variance is maximized along the principal components. The process can be summarized as follows:

**1. Mean-Centering:** Before applying PCA, the mean of each feature (time step) is subtracted from the corresponding column to center the data. The mean-centered matrix is denoted as $D_{\text{centered}}$, with each element given by:

$$D_{\text{centered}}(i, j) = D_{\text{train}}(i, j) - \bar{D}_{\text{train}}(j), \tag{1}$$

where $\bar{D}_{\text{train}}(j)$ is the mean of the $j$-th column.

**2. Covariance Matrix:** The covariance matrix $C$ is computed based on the mean-centered data:

$$C = \frac{1}{n-1} D_{\text{centered}}^T \cdot D_{\text{centered}}. \tag{2}$$

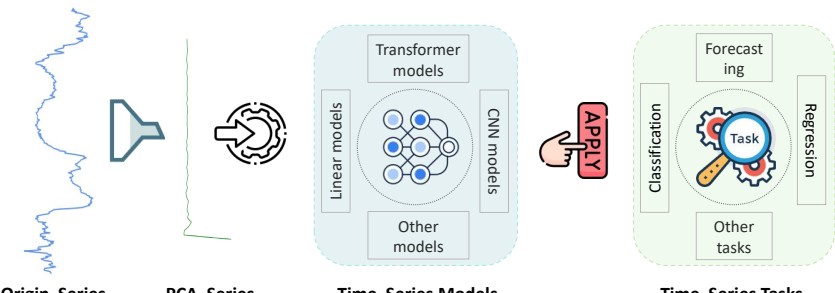

Figure 2: PCA is utilized for time series reduction in temporal dimension to enhance the efficiency of model training and inference in TSA.

**3. Eigenvalue Decomposition:** PCA involves finding the eigenvalues $\lambda_1, \lambda_2, ..., \lambda_m$ and corresponding eigenvectors $v_1, v_2, ..., v_m$ of the covariance matrix $C$. The eigenvalues represent the variance along each principal component.

**4. Selecting Principal Components:** The principal components are selected based on the proportion of variance they explain. The $k$-th principal component is given by $PC_k = D_{\text{centered}} \cdot v_k$, where $v_k$ is the $k$-th eigenvector.

**5. Reducing Dimensionality:** To reduce the dimensionality of the time series data, the original data matrix is projected onto the first $k$ principal components, forming a new matrix $D_{\text{pca}} \in \mathbb{R}^{n \times k}$:

$$D_{\text{pca}} = D_{\text{centered}} \cdot V_k, \tag{3}$$

where $V_k$ is a matrix containing the first $k$ eigenvectors as columns. $k$ s typically much smaller than $m$, thereby achieving series dimensionality reduction.

When dealing with high-dimensional and large datasets, the original PCA technique might pose computational challenges. However, optimization algorithms such as Randomized PCA (Rokhlin et al., 2010), Sparse PCA (Zou et al., 2006), and parallel computation (Andrecut, 2009) can significantly expedite the PCA computation process, making PCA preprocessing computationally efficient compared to the subsequent deep-learning model training/inference stage.

In our study, PCA is utilized in the preprocessing stage for time series dimensionality reduction before feeding the data into various deep-learning based TSA models. The original time series is transformed into a PCA series containing the top $k$ principal components. The essence of PCA guarantees that the transformed PCA series preserves the fundamental features of the original series. Prior to training the TSA model, PCA is fitted on the training set to obtain PCA-related parameters (covariance matrix, eigenvalues, and eigenvectors). During inference, each time series sample is preprocessed with the fitted PCA before being input into the TSA model. For typical time series models (Zeng et al., 2023; Zhou et al., 2021), the original historical series can be directly transformed using PCA. In contrast, for patch-based models (Nie et al., 2022) that split the series into patches, all patches from the original series are transformed separately using PCA and then concatenated. The reduced-dimensional PCA series serves as the input for the subsequent TSA model and is applied to the various downstream TSA task, as illustrated in Fig. 2. This method enables more efficient training and inference while retaining the essential information captured by the principal components. While this PCA-based preprocessing method is originally designed for univariate TSA, it can readily be extended to multivariate TSA if subsequent TSA models are channel-independent (Nie et al., 2022).

### 3.2 Intuitional justifications on PCA's Effectiveness in Time Series Reduction

PCA is effective in time series data reduction due to several key advantages. It serves as an efficient noise reduction tool by filtering out low-variance noise and retaining high-variance features. Additionally, PCA preserves the critical statistical characteristics of the original series.

**PCA acts as an efficient tool for noise reduction within historical series.** By projecting the original historical series onto a new set of orthogonal components, PCA effectively filters out the noise contained in the lower variance components, thus retaining the core information of the historical series. This noise reduction can be visualized through PCA-inverse transformations, which

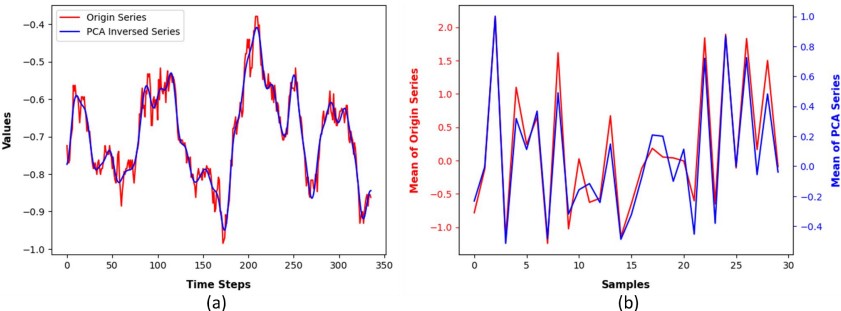

Figure 3: (a) PCA-inversed series. The PCA-inversed series is significantly smoother than the original series, indicating that PCA effectively filters out noise while preserving essential features. (b) Mean value distribution. The distributions of mean values for the original series and the PCA-reduced series show a high degree of overlap, demonstrating that PCA retains the key statistical characteristics.

reconstruct the original time series from the principal components. Fig. 3 (a) demonstrates that the inverse-transformed series is significantly smoother than the original series, indicating that PCA effectively filters out noise while preserving essential features. Consequently, the PCA process helps to mitigate overfitting in subsequent TSA models.

**PCA retains the critical statistical characteristics of the original time series data.** The key statistical characteristics preserved by PCA include the mean, sum, peak values, and higher-order moments. Specifically, for the mean/sum values of historical series, PCA simply maps the original time points into a different coordinate system, preserving the relative mean/sum values of different historical series. Fig. 3 (b) illustrates the mean values distributions of 30 original series and their corresponding PCA series, demonstrating a high degree of overlap between the two distributions. For the peak values, the distribution of peaks in the PCA series also shows a high degree of similarity to that of the original series. Furthermore, PCA preserves higher-order moments, including skewness and kurtosis (Bai & Ng, 2005), because its linear transformation ensures that these higher-order statistical characteristics remain intact. The preservation of these distinctive statistical characteristics in the PCA-reduced series enables effective learning by subsequent TSA models.

**Specific trends and periodic patterns in historical series may not be crucial for the learning of TSA models.** While some time series research focuses on extracting trends or periodicity (Zhou et al., 2022; Wu et al., 2023), we argue that these specific trends and periodic patterns are not necessarily essential for effective TSA model learning. For example, if all positive trends in the historical series are reversed, the relative distribution of the data remains unchanged, and the TSA model's ability to learn and predict is not impaired. Similarly, if the periodic components in all historical series are minified or magnified, the model's predictive capabilities should not be affected. These observations suggest that the presence of specific trend or periodicity in historical series is not necessarily essential for the learning process of TSA models. Instead, the presence of consistent and coherent patterns is sufficient for models to provide accurate predictions. Therefore, although PCA may alter the trend or periodicity, it introduces new coherent patterns—such as the main directions of variation, denoised low-dimensional representations, and latent features—that effectively benefit TSA model learning.

## 4 EXPERIMENTS

To validate the effectiveness of PCA in time series compression and temporal dimensionality reduction, experiments are conducted on three mainstream TSA tasks: time series classification (TSC), time series forecasting (TSF), and time series extrinsic regression (TSER). Table 1 summarizes the benchmarks across 13 datasets, with detailed descriptions in Appendix A. Four types of advanced time series models are evaluated for TSA. 1) MLP(linear)-based model: Linear (Zeng et al., 2023); 2) Transformer-based models: Informer (Zhou et al., 2021), FEDformer (Zhou et al., 2022), and PatchTST (Nie et al., 2022); 3) CNN-based model: TimesNet (Wu et al., 2023); 4) RNN-based models: Gated Recurrent Unit (GRU) (Chung et al., 2014) and Long Short-Term Memory (LSTM) (Hochreiter, 1997). A detailed description of these models can be found in Appendix B. Sections 4.1-4.3 compare the performance of TSA models in TSC, TSF, and TSER tasks, both with and without PCA preprocessing. The results show that PCA maintains model performance by retaining the principal information of the original time series. Section 4.4 highlights PCA's optimization

of training/inference in TSA models, notably accelerating Informer by up to 40% and reducing TimesNet's GPU memory usage by 30%.

Table 1: Overview of experiment benchmarks.

| Task | Datasets | Metrics | Series Length |
|---|---|---|---|
| Classification | EthanolConcentration, Handwriting, SelfRegulationSCP1, SelfRegulationSCP2, UWaveGestureLibrary | Accuracy | 152-1751 |
| Forecasting | ETTh1, ETTh2, ETTm1, ETTm2 | MSE, MAE | 336 |
| Regression | FloodModeling1, FloodModeling2, FloodModeling3, Covid3Month | RMSE, MAE | 84-266 |

### 4.1 TIME SERIES CLASSIFICATION

We perform sequence-level TSC experiments using five datasets from the UEA archive (Bagnall et al., 2018). Our experiments adhere to the settings outlined in the "Time Series Library" (Wu et al., 2023) [1], with the exception that we focus solely on the last dimension of each dataset, resulting in a univariate TSC problem. Model performance is assessed using the accuracy metric. Historical series lengths vary across datasets, with principal components set to 16, 48, and 96.

Table 2: TSC experiments. The accuracy metric is adopted, where higher accuracy indicates better performance. The * symbols after models indicate the application of PCA before inputting the series into the models. Bold font is the superior result. PCA preprocessing retains series principal information, matching TSC performance with original series, and enabling training/inference acceleration.

| | Linear | Linear* | Informer | Informer* | FEDformer | FEDformer* | TimesNet | TimesNet* |
|---|---|---|---|---|---|---|---|---|
| EthanolConcentration | 0.297 | **0.300** | 0.278 | **0.285** | **0.312** | 0.297 | 0.281 | **0.331** |
| Handwriting | 0.118 | **0.127** | **0.160** | 0.118 | **0.133** | 0.129 | **0.185** | 0.121 |
| SelfRegulationSCP1 | **0.884** | 0.805 | **0.846** | 0.703 | 0.556 | **0.806** | **0.918** | 0.686 |
| SelfRegulationSCP2 | 0.528 | **0.539** | 0.533 | **0.628** | 0.533 | **0.600** | 0.583 | **0.592** |
| UWaveGestureLibrary | **0.575** | 0.409 | **0.550** | 0.522 | 0.309 | **0.538** | **0.603** | 0.491 |
| Better Count | 2 | 3 | 3 | 2 | 2 | 3 | 3 | 2 |

Table 2 displays the TSC results of the Linear, Informer, FEDformer, and TimesNet models, both with and without PCA preprocessing. From the evaluation of four models across five datasets, resulting in a total of 20 metrics, PCA shows better performance in 10 metrics. These results reveal PCA's efficacy in extracting series information for TSC tasks without performance loss, enabling faster training/inference. Details on the specific acceleration effects during training/inference for various models can be found in Section 4.4.

### 4.2 TIME SERIES FORECASTING

For TSF, we follow the evaluation procedure from the study (Zhou et al., 2021), using MSE and MAE on z-score normalized data. We assess the models on four ETT datasets (Zhou et al., 2021), utilizing the "oil temperature" variable for both training and testing. To comprehensively evaluate the models, we adopt four distinct prediction lengths, specifically 96, 192, 336, and 720. The historical input series has a length of 336, and the number of principal components is set to 48.

Table 3 presents the forecasting results for different time series models. The results of Linear are adapted from the study (Zeng et al., 2023). The remaining experiments are followed the experimental setup of Time Series Library. The table illustrates that Informer performs better with PCA preprocessing, whereas FEDformer exhibits a performance decline with PCA preprocessing. For Linear and TimesNet, the performance remains largely unchanged, regardless of PCA preprocessing. These findings affirm that PCA can efficiently extract features from historical series in TSF tasks without compromising the TSA models' performance, thereby achieving accelerated training/inference. Model-specific acceleration details are in Section 4.4.

---

[1] https://github.com/thuml/Time-Series-Library

Table 3: TSF experiments. Lower MSE/MAE indicates better performance. The * symbols after models indicate the application of PCA before inputting the series into the models. Bold font represents the superior result. PCA preprocessing retains series principal information, matching TSF performance with original series, and enabling training/inference acceleration.

| Models | | Linear | | Linear* | | Informer | | Informer* | | FEDformer | | FEDformer* | | TimesNet | | TimesNet* | |
|---|---|---|---|---|---|---|---|---|---|---|---|---|---|---|---|---|---|
| Metric | | MSE | MAE | MSE | MAE | MSE | MAE | MSE | MAE | MSE | MAE | MSE | MAE | MSE | MAE | MSE | MAE |
| ETTh1 | 96 | 0.189 | 0.359 | **0.063** | **0.087** | **0.177** | **0.352** | 0.188 | 0.365 | **0.074** | **0.217** | 0.088 | 0.231 | 0.336 | 0.490 | **0.244** | **0.424** |
| | 192 | **0.078** | **0.212** | 0.086 | 0.221 | 0.191 | 0.368 | **0.096** | **0.243** | 0.080 | 0.229 | 0.097 | 0.240 | 0.139 | **0.292** | **0.137** | 0.299 |
| | 336 | **0.091** | **0.237** | **0.091** | **0.237** | 0.148 | 0.311 | **0.103** | **0.251** | 0.074 | 0.218 | 0.081 | 0.227 | **0.220** | **0.398** | 0.252 | 0.432 |
| | 720 | **0.172** | **0.340** | 0.190 | 0.361 | 0.268 | 0.444 | **0.143** | **0.304** | 0.079 | 0.226 | 0.224 | 0.399 | 0.328 | 0.504 | **0.268** | **0.449** |
| ETTh2 | 96 | **0.133** | **0.283** | 0.134 | **0.283** | 0.286 | 0.426 | **0.220** | **0.367** | **0.148** | **0.309** | 0.164 | 0.319 | 0.162 | **0.313** | **0.161** | 0.320 |
| | 192 | **0.176** | **0.330** | 0.180 | 0.335 | **0.209** | **0.373** | 0.234 | 0.387 | **0.175** | **0.341** | 0.192 | 0.348 | 0.215 | 0.368 | **0.151** | **0.309** |
| | 336 | 0.213 | 0.371 | **0.201** | **0.362** | 0.317 | 0.466 | **0.258** | **0.409** | **0.194** | 0.358 | 0.198 | **0.356** | 0.233 | 0.394 | 0.333 | 0.470 |
| | 720 | **0.292** | **0.440** | 0.366 | 0.497 | **0.405** | **0.520** | 0.429 | 0.535 | **0.253** | **0.412** | 0.324 | 0.464 | 0.352 | 0.476 | **0.329** | **0.467** |
| ETTm1 | 96 | **0.028** | **0.125** | 0.029 | 0.126 | 0.176 | 0.368 | **0.052** | **0.172** | 0.071 | 0.212 | **0.038** | **0.148** | **0.074** | **0.216** | 0.144 | 0.320 |
| | 192 | 0.043 | 0.154 | **0.042** | **0.151** | 0.129 | 0.289 | **0.106** | **0.256** | 0.059 | 0.185 | 0.063 | 0.194 | 0.191 | 0.367 | **0.172** | **0.349** |
| | 336 | 0.059 | 0.180 | **0.056** | **0.176** | 0.156 | 0.323 | **0.150** | **0.310** | 0.063 | 0.197 | 0.113 | 0.259 | 0.200 | 0.368 | **0.183** | **0.356** |
| | 720 | **0.080** | **0.211** | 0.081 | 0.212 | 0.232 | 0.403 | **0.157** | **0.320** | 0.079 | 0.221 | 0.122 | 0.271 | **0.223** | **0.400** | 0.224 | 0.403 |
| ETTm2 | 96 | 0.066 | 0.189 | **0.065** | **0.188** | **0.075** | **0.209** | 0.090 | 0.229 | 0.141 | 0.299 | **0.077** | **0.212** | **0.083** | **0.218** | 0.105 | 0.250 |
| | 192 | 0.094 | 0.230 | **0.092** | **0.228** | 0.126 | 0.280 | **0.117** | **0.265** | 0.126 | 0.275 | **0.106** | **0.249** | 0.126 | 0.273 | 0.154 | 0.312 |
| | 336 | **0.120** | **0.263** | 0.123 | 0.267 | 0.163 | 0.322 | **0.160** | **0.317** | 0.164 | 0.316 | **0.138** | **0.286** | 0.168 | 0.326 | 0.173 | 0.335 |
| | 720 | 0.175 | **0.320** | **0.174** | **0.320** | 0.231 | 0.383 | 0.248 | 0.398 | 0.176 | 0.324 | **0.168** | **0.317** | **0.170** | 0.332 | **0.170** | **0.331** |
| Better Count | | 19 | | 17 | | 10 | | 22 | | 21 | | 11 | | 17 | | 16 | |

For the patch-based model PatchTST (Nie et al., 2022), PCA preprocessing is performed separately on each patch series, where each patch has a length of 16 and is reduced to 2 through PCA. Subsequently, all PCA subseries are concatenated together and fed into the backbone of PatchTST. PCA can also accelerate the training/inference of PatchTST, and its detailed TSF results are shown in Appendix C. Furthermore, although RNN-based models are less prevalent in TSA, to more comprehensively evaluate the impact of PCA preprocessing, two RNN-based models are assessed: GRU and LSTM. The results show that PCA preprocessing maintains the predictive performance of these models while providing greater acceleration during training and inference. The detailed TSF results for RNN-based models are shown in Appendix D.

## 4.3 TIME SERIES EXTRINSIC REGRESSION

We conduct TSER experiments using four univariate datasets from the study (Tan et al., 2021). These datasets are from the domains of environmental monitoring and disease diagnosis. The metrics RMSE and MAE are used to evaluate the performance of the models. The length of historical series varies across datasets, with a length of 266 for FloodModeling datasets and 84 for the Covid3Month dataset. The number of principal components is set to 48 for FloodModeling and 16 for Covid3Month.

Table 4: TSER experiments. Lower RMSE/MAE indicates better performance. The * symbols after models indicate the application of PCA before inputting the series into the models. Bold font represents the superior result. PCA preprocessing retains series principal information, matching TSER performance with original series, and enabling training/inference acceleration.

| Models | Linear | | Linear* | | Informer | | Informer* | | FEDformer | | FEDformer* | | TimesNet | | TimesNet* | |
|---|---|---|---|---|---|---|---|---|---|---|---|---|---|---|---|---|
| Metric | RMSE | MAE | RMSE | MAE | RMSE | MAE | RMSE | MAE | RMSE | MAE | RMSE | MAE | RMSE | MAE | RMSE | MAE |
| FloodModeling1 | 0.132 | 0.032 | **0.019** | **0.015** | **0.019** | **0.015** | 0.019 | 0.015 | 0.024 | 0.019 | **0.020** | **0.015** | 0.019 | 0.015 | 0.019 | 0.015 |
| FloodModeling2 | 0.030 | 0.016 | 0.021 | 0.009 | **0.019** | **0.006** | 0.019 | 0.007 | 0.021 | **0.008** | 0.020 | 0.009 | 0.018 | 0.006 | 0.019 | 0.006 |
| FloodModeling3 | 0.047 | 0.020 | **0.023** | **0.017** | 0.024 | 0.020 | **0.023** | **0.018** | 0.033 | 0.026 | **0.031** | **0.023** | 0.023 | 0.017 | 0.023 | 0.017 |
| Covid3Month | 0.116 | 0.069 | **0.045** | **0.034** | **0.043** | 0.034 | 0.045 | **0.033** | 0.063 | 0.041 | **0.045** | **0.034** | 0.045 | 0.035 | 0.045 | 0.035 |
| Better Count | 0 | | 8 | | 5 | | 6 | | 1 | | 7 | | 8 | | 7 | |

Table 4 indicates that Linear and FEDformer exhibit improved performance with PCA preprocessing in TSER, as evidenced by lower RMSE and MAE scores. In contrast, the performance of the Informer and TimesNet models is almost unaffected by the use of PCA preprocessing. These results illustrate that PCA extracts series features efficiently in TSER, sustains TSA models' performance, and speeds up training/inference, as detailed in Section 4.4.

## 4.4 TRAINING/INFERENCE OPTIMIZATION WITH PCA

The aforementioned results indicate that PCA successfully preserves essential information in time series while maintaining TSA models' performance across tasks of TSC, TSF, and TSER, and we also investigate the impact of PCA on reducing computational burden and accelerating training/inference.

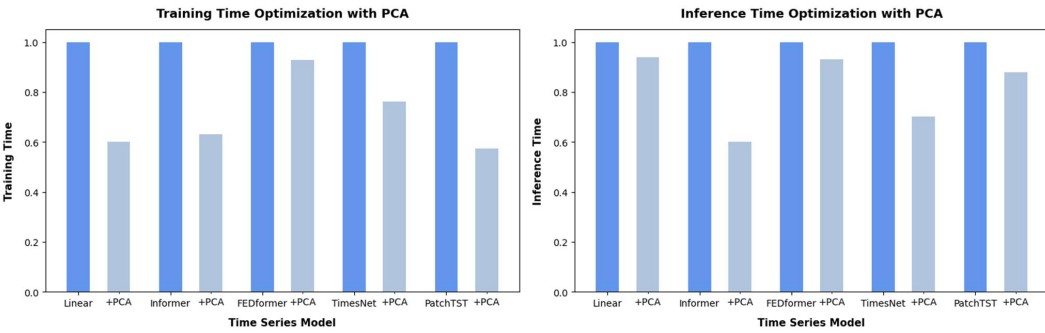

Figure 4: Training/inference time of various time series models with and without PCA preprocessing.

Fig. 4 illustrates the average training/inference time of various time series models with and without PCA preprocessing across the three tasks. The running time of each model is normalized to facilitate comparison. Since the time consumed by PCA preprocessing is negligible compared to the model training/inference, it is not included in the figure, and the detailed results (including the time taken by PCA processing) are presented in Appendix E. Results show significant training time acceleration for Linear, Informer, and PatchTST, with up to 40% improvement. Additionally, PCA preprocessing leads to varying degrees of acceleration in the model's inference process. TimesNet shows 30% acceleration for both training and inference, while FEDformer shows 10% improvement.

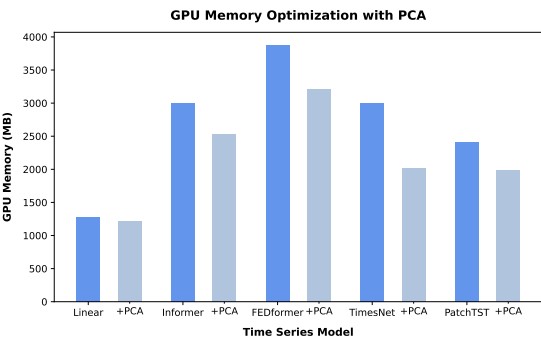

Figure 5: GPU memory utilization of various time series models with and without PCA preprocessing.

Fig. 5 illustrates the impact of PCA preprocessing on the average GPU memory utilization of various time series models across three tasks. The results demonstrate that PCA preprocessing can significantly reduce GPU memory usage for some models. Specifically, for TimesNet, PCA preprocessing leads to a 30% reduction in GPU memory usage. For Informer, FEDformer, and PatchTST, the reduction is approximately 15%. However, for the Linear model, there is almost no reduction in GPU memory usage. These findings suggest that PCA preprocessing can be an effective method for reducing the computational burden and accelerating training and inference in time series models, but its impact varies depending on the specific model and scenario.

## 5 COMPARISON OF PCA WITH OTHER HISTORICAL INPUT SERIES REDUCTION METHODS

To further confirm the effectiveness of PCA as a reduction method for time series in temporal dimension, a comparison is made between PCA and other input series reduction techniques in TSF tasks, such as directly shortening the length of the historical series to 48, and downsampling every 7 time steps to reduce the length to 48. Table 5 indicates that both direct input shortening and downsampling significantly compromise performance of Linear, with direct input shortening having a more substantial impact.

Furthermore, we conduct experiments of adding linear/1D-CNN dimension reduction layer at the beginning of the deep-learning models to automatically compress the original series before subsequent

Table 5: The comparison of PCA, shortening and downsampling as series reduction methods. Bold font represents the superior result.

| Models | | Linear | | PCA | | Shortening | | Downsampling | |
|---|---|---|---|---|---|---|---|---|---|
| Metric | | MSE | MAE | MSE | MAE | MSE | MAE | MSE | MAE |
| ETTm1 | 96 | **0.028** | **0.125** | 0.029 | 0.126 | 0.033 | 0.131 | 0.030 | 0.128 |
| | 192 | 0.043 | 0.154 | **0.042** | **0.151** | 0.052 | 0.167 | 0.045 | 0.155 |
| | 336 | 0.059 | 0.180 | **0.056** | **0.176** | 0.073 | 0.199 | 0.060 | 0.179 |
| | 720 | **0.080** | **0.211** | 0.081 | 0.212 | 0.103 | 0.240 | 0.084 | 0.215 |
| ETTm2 | 96 | 0.066 | 0.189 | **0.065** | **0.188** | 0.081 | 0.208 | 0.068 | 0.194 |
| | 192 | 0.094 | 0.230 | **0.092** | **0.228** | 0.114 | 0.251 | 0.095 | 0.233 |
| | 336 | **0.120** | **0.263** | 0.123 | 0.267 | 0.148 | 0.292 | 0.124 | 0.268 |
| | 720 | 0.175 | **0.320** | **0.174** | **0.320** | 0.203 | 0.348 | 0.179 | 0.325 |
| Better Count | | 7 | | 10 | | 0 | | 0 | |

computations. This increases the complexity of the models, potentially exacerbating the inherent overfitting issues in time series modeling. Table 6 shows that incorporating a linear/1D-CNN dimension reduction layer (denoted as "Model+/Model++" in the table) is notably inferior to PCA-based reduction (denoted as "Model*" in the table), and also inferior to models without such layers (refer to Table 3). Moreover, updating the parameters of these reduction layers during each training iteration can diminish the efficiency gains and memory advantages. These experiments highlight PCA's effectiveness as a reduction technique of historical input series, maintaining models' performance while reducing temporal dimensionality.

Table 6: The comparison of PCA-based reduction and incorporation of a linear/1D-CNN dimension reduction layer. Bold font represents the best result.

| Models | | Linear* | | Linear+ | | Linear++ | | Informer* | | Informer+ | | Informer++ | |
|---|---|---|---|---|---|---|---|---|---|---|---|---|---|---|
| Metric | | MSE | MAE | MSE | MAE | MSE | MAE | MSE | MAE | MSE | MAE | MSE | MAE |
| ETTm1 | 96 | **0.029** | **0.126** | 0.030 | 0.129 | 0.031 | 0.132 | **0.052** | **0.172** | 0.120 | 0.284 | 0.059 | 0.192 |
| | 192 | **0.042** | **0.151** | 0.044 | 0.155 | **0.042** | 0.153 | **0.106** | **0.256** | 0.185 | 0.367 | 0.163 | 0.344 |
| | 336 | **0.056** | **0.176** | 0.065 | 0.189 | 0.060 | 0.182 | **0.150** | **0.310** | 0.153 | 0.314 | 0.213 | 0.390 |
| | 720 | **0.081** | **0.212** | 0.083 | 0.214 | 0.083 | 0.215 | 0.157 | 0.320 | 0.220 | 0.394 | **0.138** | **0.300** |
| ETTm2 | 96 | **0.065** | **0.188** | 0.066 | 0.190 | 0.071 | 0.199 | **0.090** | **0.229** | **0.090** | 0.232 | 0.094 | 0.236 |
| | 192 | **0.092** | **0.228** | 0.096 | 0.233 | 0.097 | 0.236 | **0.117** | **0.265** | 0.147 | 0.302 | 0.182 | 0.345 |
| | 336 | 0.123 | **0.267** | **0.122** | **0.267** | 0.126 | 0.272 | **0.160** | **0.317** | 0.237 | 0.385 | 0.195 | 0.351 |
| | 720 | **0.174** | **0.320** | 0.177 | 0.323 | 0.177 | 0.324 | **0.248** | **0.398** | 0.258 | 0.403 | 0.265 | 0.417 |
| Better Count | | 15 | | 2 | | 1 | | 14 | | 1 | | 2 | |

# 6 CONCLUSION

Our study challenges the perception that PCA, by disrupting sequential relationships in time series, is unsuitable for TSA. Instead, we find it efficient for handling TSA tasks. PCA is innovatively applied to achieve temporal dimensionality reduction while safeguarding essential information within time series. Its effect is evaluated on four types of advanced time series models, namely Linear, Transformer, CNN and RNN models, across three typical TSA tasks: classification, forecasting, and regression. The results show that PCA reduces computational burden without compromising performance. Specifically, in TSC, the performance with PCA is better in 50.0% of cases; in TSF, 49.6% of cases; and in TSRE, 66.7%. Notably, PCA accelerates Informer's training and inference by up to 40%, with a minimum of 10% speedup for other models. Additionally, PCA reduces GPU memory usage by 15% for Transformer-based models and 30% for CNN-based models. The study discusses PCA's theoretical effectiveness in denoising and preserving statistical information, further substantiating its superiority over alternative dimensionality reduction methods such as series shortening, downsampling, and the integration of additional reduction layers.

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

# Supplemental Materials for "Revisiting PCA for Time Series Reduction in Temporal Dimension"

## A  DATA DESCRIPTION

The experimental data comprises 13 widely-used datasets from various domains, each distinguished by unique attributes:

- ETT (Zhou et al., 2021): The ETT dataset includes two hourly-level datasets (ETTh1 and ETTh2) and two 15-minute-level datasets (ETTm1 and ETTm2). Each dataset comprises seven oil and load features of electricity transformers spanning from July 2016 to July 2018. There are 7 variables for each dataset, with 17,420 time steps for ETTh and 69,680 time steps for ETTm. The series in these datasets exhibit strong periodicity. For univariate forecasting, only the "oil temperature" variable is used for training and testing.

- EthanolConcentration (Bagnall et al., 2018): The dataset comprises 544 time series formed by the raw spectra of water and ethanol solutions in authentic whisky bottles, with each series having a length of 1,751. Ethanol concentrations range from 35%, 38%, 40%, to 45%. The primary objective of this dataset is to ascertain the ethanol concentration (category) within each sample. As a multivariate dataset, each variable corresponds to measurements at different wavelengths, spanning Ultraviolet (UV) light, Visible (VIS) light, and Near Infrared (NIR). For our experiments, the NIR variable is selected.

- Handwriting (Bagnall et al., 2018): This dataset comprises 1,000 time series samples of subjects wearing a smartwatch while writing the 26 English letters. Each series has a length of 152, with three dimensions corresponding to three accelerometer values. In our experiments, we select the last dimension.

- SelfRegulationSCP (Bagnall et al., 2018): SelfRegulationSCP encompasses two datasets, SelfRegulationSCP1 and SelfRegulationSCP2, involving self-regulation of slow cortical potentials. In SelfRegulationSCP1, data from a healthy subject include cursor movement on a computer screen, with visual feedback regulating slow cortical potentials (Cz-Mastoids). SelfRegulationSCP1 consists of 561 series samples, each with a length of 896. In Self-RegulationSCP2, data from an artificially respirated ALS patient similarly involve cursor movement, with auditory and visual feedback regulating slow cortical potentials. SelfRegu-lationSCP2 comprises 380 series samples, each with a length of 1,152. The classification objective is to categorize based on recorded slow cortical potentials, where positive and negative potentials correspond to different classes. The analysis in both datasets focuses on the last dimension of the data in experiments.

- UWaveGestureLibrary (Bagnall et al., 2018): The UWaveGestureLibrary dataset comprises eight simple gestures generated from accelerometers, totaling 4,479 series samples. Each sample includes the x, y, z coordinates of a gesture, with each series having a length of 315. In the experiments, the analysis is focused on the z-coordinate series.

- FloodModeling (Tan et al., 2021): FloodModeling comprises three hourly datasets (Flood-Modeling1, FloodModeling2, and FloodModeling3). These datasets aim to predict the maximum water depth for flood modeling. The three datasets contain 673, 559, and 613 hourly rainfall events time series, respectively. Each time series in the datasets has a length of 266 time steps. These time series are utilized to predict the maximum water depth of a domain represented by a Digital Elevation Model (DEM). Both the rainfall events and DEM are synthetically generated by researchers at Monash University.

- Covid3Month (Tan et al., 2021): The Covid3Month dataset comprises 201 time series, where each time series represents the daily confirmed cases for a country. The length of each time series is 84. The objective of this dataset is to predict the COVID-19 death rate on April 1, 2020, for each country using the daily confirmed cases over the preceding three months.

## B  DETAILS ON TIME SERIES MODELS

The descriptions and implementations of the evaluated time series models are provided below:

Linear (Zeng et al., 2023): The Linear model represents a groundbreaking method utilizing a linear model, outperforming a substantial portion of Transformer-based models for TSF. The corresponding code is accessible at: `https://github.com/cure-lab/LTSF-Linear`.

Informer (Zhou et al., 2021): Informer is an efficient Transformer architecture specifically designed for TSF. The code for this model can be found at `https://github.com/zhouhaoyi/Informer2020`.

FEDformer (Zhou et al., 2022): FEDformer is an efficient Transformer architecture that reduces computational complexity through frequency-domain self-attention, utilizing Fourier or wavelet transforms and random selection of frequency bases. The code for this model can be accessed at: `https://github.com/MAZiqing/FEDformer`.

TimesNet (Wu et al., 2023): TimesNet transforms 1D time series into a set of 2D tensors based on multiple periods and utilizes a CNN-based model to extract features. The code for TimesNet can be found at `https://github.com/thuml/Time-Series-Library`.

PatchTST (Nie et al., 2022): PatchTST employs a segmentation approach for time series by dividing it into multiple time patches, treating each as a token. The model uses an attention module to learn the relationships between these tokens. The publicly available source code for PatchTST can be found at `https://github.com/yuqinie98/patchtst`.

## C    TSF RESULTS OF PATCHTST WITH PCA PREPROCESSING

PCA preprocessing is separately applied to each patch series in the patch-based time series model PatchTST. Additionally, to enhance prediction stability, PatchTST employs instance normalization technology (Kim et al., 2022). However, integrating this technology with PCA series poses challenges: the fluctuation of PCA series is considerable, and adding instance normalization further destabilizes the predictions. Consequently, after applying PCA processing, we exclude the instance normalization module from PatchTST. For comparative analysis, we also assess the performance of PatchTST without the instance normalization module on the original series.

Table 7 presents the forecasting results of PatchTST. It is observed that the original PatchTST achieves optimal performance. However, a surprising discovery is the pivotal role played by the instance normalization process in PatchTST. Omitting the instance normalization module results in a significant deterioration in PatchTST performance, exhibiting much worse results compared to training PatchTST (also without the instance normalization module) after PCA preprocessing. These findings suggest that PCA is effective for patch-based time series models, yet further exploration is required to identify alternative methods to instance normalization.

## D    TSF RESULTS OF RNN-BASED MODELS WITH PCA PREPROCESSING

Due to issues with gradient vanishing or exploding (Hanin, 2018), RNN-based models exhibit unstable performance in TSA with long historical series windows and have consequently been increasingly supplanted by Transformer, linear, and CNN-based models. Nonetheless, to more comprehensively evaluate the impact of PCA preprocessing, we assess its effect on RNN-based models for TSF tasks. Specifically, two typical RNN-based models, GRU (Chung et al., 2014) and LSTM (Hochreiter, 1997), are tested. Original historical series or PCA series are fed into the GRU or LSTM cells to extract features, and their hidden state $h$, containing the feature information, are projected and transformed to obtain the final predictions. Table 8 shows that for GRU, PCA preprocessing leads to superior performance in 18 out of 32 settings, and for LSTM, PCA preprocessing achieves better results in half of the settings. These results indicate that PCA preprocessing does not degrade the performance of RNN-based models. Additionally, since RNN models process time series sequentially, their computational cost is more sensitive to the length of the model input. Table 9 demonstrates that PCA preprocessing has a significant acceleration effect on RNN-based models, reducing training time to one-fourth and inference time to one-third of the original times. Although RNN-based models are not as commonly used as other models, PCA remains an effective tool for time series reduction in scenarios where they are appropriate.

Table 7: TSF experiments of PatchTST. The - symbol after the model signifies the removal of instance normalization processing, and the * symbol after the model indicates the application of PCA. The best result is indicated in bold font, while the second-best result is underlined.

| Models | | PatchTST | | PatchTST- | | PatchTST* | |
|---|---|---|---|---|---|---|---|
| Metric | | MSE | MAE | MSE | MAE | MSE | MAE |
| ETTh1 | 96 | **0.055** | **0.179** | 0.141 | 0.300 | 0.073 | 0.214 |
| | 192 | **0.071** | **0.205** | 0.196 | 0.368 | 0.082 | 0.234 |
| | 336 | **0.081** | **0.225** | 0.186 | 0.360 | 0.087 | 0.237 |
| | 720 | **0.087** | **0.232** | 0.372 | 0.527 | 0.131 | 0.289 |
| ETTh2 | 96 | **0.129** | **0.282** | 0.232 | 0.381 | 0.166 | 0.324 |
| | 192 | **0.168** | **0.328** | 0.221 | 0.368 | 0.214 | 0.376 |
| | 336 | **0.185** | **0.351** | 0.537 | 0.542 | 0.224 | 0.390 |
| | 720 | **0.224** | **0.383** | 0.485 | 0.561 | 0.298 | 0.447 |
| ETTm1 | 96 | **0.026** | **0.121** | 0.122 | 0.296 | 0.031 | 0.134 |
| | 192 | **0.039** | **0.150** | 0.127 | 0.299 | 0.041 | 0.157 |
| | 336 | **0.053** | **0.173** | 0.252 | 0.450 | 0.058 | 0.184 |
| | 720 | **0.074** | **0.207** | 0.276 | 0.454 | 0.084 | 0.220 |
| ETTm2 | 96 | **0.065** | **0.186** | 0.130 | 0.281 | 0.070 | 0.200 |
| | 192 | **0.094** | **0.231** | 0.132 | 0.283 | 0.098 | 0.238 |
| | 336 | **0.120** | **0.265** | 0.165 | 0.322 | 0.124 | 0.269 |
| | 720 | **0.171** | **0.322** | 0.286 | 0.424 | 0.177 | 0.328 |

Table 8: TSF Results of RNN-based models. The * symbols after models indicate the application of PCA before inputting the series into the models. Bold font represents the superior result.

| Models | | GRU | | GRU* | | LSTM | | LSTM* | |
|---|---|---|---|---|---|---|---|---|---|
| Metric | | MSE | MAE | MSE | MAE | MSE | MAE | MSE | MAE |
| ETTh1 | 96 | 0.182 | 0.349 | **0.167** | **0.141** | 0.323 | 0.498 | **0.209** | **0.382** |
| | 192 | 0.326 | 0.487 | **0.148** | **0.316** | 0.354 | 0.515 | **0.292** | **0.469** |
| | 336 | 0.233 | 0.408 | **0.144** | **0.310** | 0.387 | 0.553 | **0.261** | **0.441** |
| | 720 | 0.266 | 0.441 | **0.183** | **0.352** | **0.370** | **0.539** | 1.565 | 1.215 |
| ETTh2 | 96 | 0.307 | 0.405 | **0.257** | **0.403** | **0.153** | **0.313** | 0.419 | 0.516 |
| | 192 | 0.227 | 0.382 | 0.279 | 0.417 | **0.207** | **0.364** | 0.298 | 0.440 |
| | 336 | 0.320 | 0.462 | **0.273** | **0.419** | 0.333 | 0.461 | **0.249** | **0.497** |
| | 720 | 0.392 | 0.502 | **0.285** | **0.435** | 0.421 | 0.534 | **0.349** | **0.378** |
| ETTm1 | 96 | **0.070** | **0.198** | 0.164 | 0.335 | **0.091** | **0.249** | 0.130 | 0.292 |
| | 192 | **0.141** | **0.295** | 0.188 | 0.360 | 0.175 | 0.349 | **0.131** | **0.280** |
| | 336 | **0.227** | **0.393** | 0.275 | 0.455 | **0.217** | **0.381** | 0.282 | 0.461 |
| | 720 | 0.400 | 0.547 | **0.268** | **0.446** | 0.368 | 0.525 | **0.289** | **0.467** |
| ETTm2 | 96 | **0.074** | **0.200** | 0.141 | 0.298 | **0.086** | **0.218** | 0.151 | 0.310 |
| | 192 | **0.119** | **0.267** | 0.187 | 0.352 | **0.119** | **0.270** | 0.211 | 0.368 |
| | 336 | 0.193 | 0.360 | **0.161** | **0.310** | 0.218 | 0.378 | **0.158** | **0.314** |
| | 720 | **0.224** | **0.368** | 0.258 | 0.408 | **0.240** | **0.385** | 0.298 | 0.446 |
| Better Count | | 14 | | 18 | | 16 | | 16 | |

Table 9: Average training/inference time (s) of RNN-based models on TSF tasks. The * symbols after the time series models indicate the application of PCA. Bold font represents the superior result.

| | GRU | GRU* | LSTM | LSTM* |
|---|---|---|---|---|
| Training time | 167.65 | **41.50** | 177.12 | **46.59** |
| PCA time | - | 0.88 | - | 0.88 |
| Inference time | 3.02 | **0.97** | 3.06 | **0.99** |
| PCA time | - | 0.01 | - | 0.01 |

# E    DETAILED TRAINING/INFERENCE TIME

Table 10 presents the average training and inference time (including PCA processing time) for various time series models, evaluated across different TSA tasks. With the assistance of PCA preprocessing, the training and inference of the models are accelerated to varying degrees.

Table 10: Average training/inference time (s) of different time series models across different TSA tasks. The * symbols after the time series models indicate the application of PCA before inputting the series into the models.

|  | Linear | Linear* | Informer | Informer* | FEDformer | FEDformer* | TimesNet | TimesNet* | PatchTST | PatchTST* |
|---|---|---|---|---|---|---|---|---|---|---|
| Training time | 25.47 | **14.82** | 336.74 | **232.16** | 1560.67 | **1450.31** | 488.65 | **372.66** | 118.04 | **67.67** |
| PCA time | - | 0.88 | - | 0.88 | - | 0.88 | - | 0.88 | - | 0.88 |
| Inference time | 0.67 | **0.63** | 4.94 | **2.97** | 12.15 | **11.31** | 5.75 | **4.03** | 1.41 | **1.24** |
| PCA time | - | 0.01 | - | 0.01 | - | 0.01 | - | 0.01 | - | 0.01 |

# F    IMPACT OF THE NUMBER OF PRINCIPAL COMPONENTS

The number of principal components is a crucial hyperparameter in PCA. If too many principal components are selected, the reduction in dimensionality may be insufficient, failing to achieve the desired acceleration in training/inference. Conversely, too few principal components can result in the loss of important features, leading to a decline in model performance.

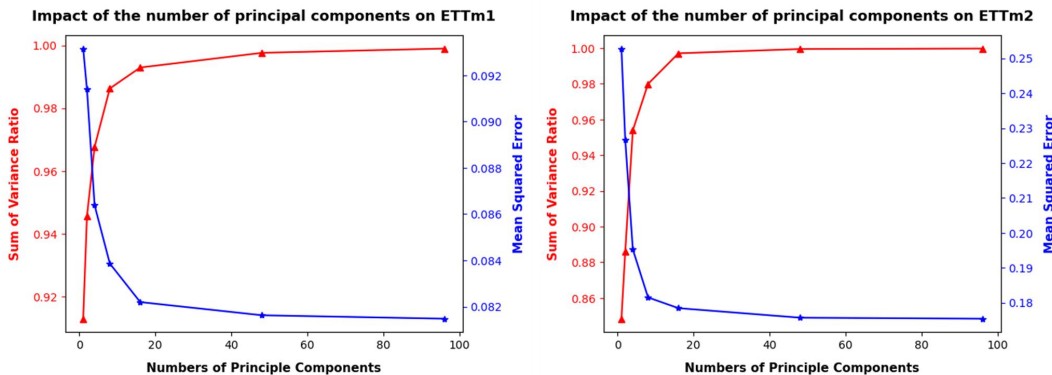

Figure 6: Impact of the number of principal components on model's performance.

Fig. 6 illustrates the impact of the number of principal components on the performance of Linear for the ETTm1 and ETTm2 datasets. The red line depicts the variation of the sum of variance ratio with the number of principal components, representing the importance of the features after PCA dimensionality reduction. As the number of principal components increases, the importance of the selected features also increases, but the rate of increase diminishes. Notably, even with only one principal component, the importance of the features is already approximately 90%, and after the number of principal components reaching to 48 (the number chosen in our experiment), further increasing the number of principal components results in minimal change in feature importance. The blue line represents the MSE of the model on the test set as a function of the number of principal components. As the number of principal components increases, the MSE decreases, but the rate of decrease also diminishes. These results suggest that selecting 48 principal components strikes a judicious balance between computational efficiency and predictive performance for TSF.

# G    COMPARISON OF PCA WITH FFT AND DWT

FFT (Duhamel & Vetterli, 1990) and DWT (Sundararajan, 2016) could also be used for temporal dimensionality reduction in time series data. In the experiments comparing PCA with FFT and DWT, the original series is first transformed from the time domain to the frequency domain using either FFT or DWT. The top k frequency components (where k is 48, the same as the number of principle components) are then selected and input these into the time series models. The results are shown

in Table 11. It is evident that the top k frequency components obtained using FFT or DWT fail to accurately capture the key information in the original series and effectively compress the series, leading to a significant decrease in model performance.

Table 11: Comparison of PCA with FFT and DWT as series reduction methods. Bold font represents the superior result.

| Models | | Linear | | PCA | | FFT | | DWT | |
|---|---|---|---|---|---|---|---|---|---|
| Metric | | MSE | MAE | MSE | MAE | MSE | MAE | MSE | MAE |
| ETTm1 | 96 | **0.028** | **0.125** | 0.029 | 0.126 | 2.110 | 1.328 | 1.827 | 1.299 |
| | 192 | 0.043 | 0.154 | **0.042** | **0.151** | 2.086 | 1.318 | 1.943 | 1.344 |
| | 336 | 0.059 | 0.180 | **0.056** | **0.176** | 2.205 | 1.356 | 1.767 | 1.279 |
| | 720 | **0.080** | **0.211** | 0.081 | 0.212 | 2.232 | 1.348 | 1.981 | 1.358 |
| ETTm2 | 96 | 0.066 | 0.189 | **0.065** | **0.188** | 3.417 | 1.467 | 1.330 | 1.010 |
| | 192 | 0.094 | 0.230 | **0.092** | **0.228** | 3.883 | 1.566 | 1.460 | 1.068 |
| | 336 | **0.120** | **0.263** | 0.123 | 0.267 | 3.273 | 1.442 | 1.421 | 1.049 |
| | 720 | 0.175 | **0.320** | **0.174** | **0.320** | 3.371 | 1.465 | 1.572 | 1.111 |
| Better Count | | 7 | | 10 | | 0 | | 0 | |

## H  TIME SERIES CLASSIFICATION IN UCR DATASETS

The UCR dataset (Dau et al., 2019) contains many time series classification datasets. To comprehensively evaluate the performance of PCA in TSC tasks, five datasets from the UCR dataset are selected for testing, as shown in Table 12. The results demonstrate that PCA preprocessing retains the principal information of the series on the UCR dataset, matches the TSC performance of the original series, and enables faster training and inference.

Table 12: TSC experiments on the UCR datasets. The accuracy metric is adopted. The * symbols after models indicate the application of PCA before inputting the series into the models. Bold font is the superior result. PCA preprocessing retains series principal information, matching TSC performance with original series, and enabling training/inference acceleration.

| | Linear | Linear* | Informer | Informer* | FEDformer | FEDformer* |
|---|---|---|---|---|---|---|
| GunPointAgeSpan | 0.864 | **0.892** | 0.889 | **0.930** | 0.775 | **0.892** |
| GunPointMaleVersusFemale | 0.731 | **0.991** | **0.997** | **0.997** | 0.706 | **0.991** |
| GestureMidAirD1 | 0.477 | **0.500** | 0.431 | **0.515** | **0.692** | 0.500 |
| GestureMidAirD2 | **0.485** | 0.454 | **0.523** | 0.400 | 0.346 | **0.415** |
| GestureMidAirD3 | **0.323** | 0.254 | **0.377** | 0.277 | 0.231 | **0.292** |
| AllGestureWiimoteX | **0.296** | 0.283 | 0.289 | **0.403** | **0.460** | 0.384 |
| AllGestureWiimoteY | 0.319 | **0.324** | **0.516** | 0.387 | 0.409 | **0.424** |
| AllGestureWiimoteZ | **0.320** | **0.320** | 0.296 | **0.372** | **0.480** | 0.366 |
| FordA | 0.504 | **0.507** | 0.523 | **0.817** | 0.639 | **0.822** |
| FordB | 0.532 | **0.546** | 0.549 | **0.709** | 0.672 | **0.685** |
| Better Count | 4 | 7 | 4 | 7 | 3 | 7 |

## I  COMPARISON OF PCA WITH REPRESENTATION LEARNING-BASED METHODS

Some representation learning-based methods, such as TS2Vec (Yue et al., 2021), T-Loss (Franceschi et al., 2019), and TimeVQVAE (Lee et al., 2023), can also compress time series data by learning their representations and then use downstream classifiers or regressors for classification or forecasting. We compared PCA with these representation learning-based methods on classification tasks. As shown in Table 13, the Linear + PCA model achieved the best performance in most settings. Additionally, it is worth noting that these representation learning-based methods are not pluggable, general-purpose

approaches and cannot be easily integrated with arbitrary time series models or tasks. Furthermore, the primary objective of these methods is to learn better representations rather than to accelerate training. As a result, they do not optimize for training efficiency or memory usage as extensively as PCA does, as shown in Table 14.

Table 13: TSC experiments of T-Loss, TS2Vec, and TimeVQVAE. The accuracy metric is adopted. Bold font is the superior result.

|  | Linear+PCA | T-Loss | TS2Vec | TimeVQVAE |
|---|---|---|---|---|
| EthanolConcentration | **0.300** | 0.289 | 0.287 | 0.203 |
| Handwriting | 0.127 | 0.255 | **0.397** | 0.218 |
| SelfRegulationSCP1 | **0.805** | 0.780 | 0.795 | 0.719 |
| SelfRegulationSCP2 | **0.539** | 0.511 | 0.525 | 0.527 |
| UWaveGestureLibrary | 0.409 | 0.622 | 0.666 | **0.668** |
| Better Count | 3 | 0 | 1 | 1 |

Table 14: Computational efficiency, and memory usage comparation of T-Loss, TS2Vec, and TimeVQ-VAE. Bold font is the superior result.

|  | Linear+PCA | T-Loss | TS2Vec | TimeVQVAE |
|---|---|---|---|---|
| Training time (s) | **14.82** | 302.50 | 25.92 | 62.98 |
| Inference time (s) | **0.59** | 2.01 | 1.65 | 68.10 |
| Memory usage (MiB) | **484** | 1290 | 2424 | 2870 |

## J  PCA'S APPLICATIONS IN ADDITIONAL TSC MODELS

Some effective specialized TSC models, such as InceptionTime (Ismail Fawaz et al., 2020) and ResNet (Cheng et al., 2021), have been developed and widely applied in various TSC tasks. We also applied PCA to these models. The results in Table 15 show that PCA is model-agnostic and remains effective even when applied to these specialized TSC models.

Table 15: TSC experiments of Inception and ResNet. The accuracy metric is adopted. The * symbols after models indicate the application of PCA before inputting the series into the models. Bold font is the superior result. PCA preprocessing retains series principal information, matching TSC performance with original series, and enabling training/inference acceleration.

|  | Inception | Inception* | ResNet | ResNet* |
|---|---|---|---|---|
| EthanolConcentration | 0.259 | **0.300** | 0.281 | **0.308** |
| Handwriting | 0.075 | **0.119** | 0.076 | **0.105** |
| SelfRegulationSCP1 | **0.833** | 0.758 | **0.867** | 0.754 |
| SelfRegulationSCP2 | 0.489 | **0.561** | 0.528 | **0.539** |
| UWaveGestureLibrary | **0.522** | 0.516 | **0.528** | 0.419 |
| Better Count | 2 | 3 | 2 | 3 |

## K  PCA'S TESTS ON ELECTRICITY AND TRAFFIC DATASETS

We also applied PCA to the commonly used TSF datasets, Electricity and Traffic. The results in Table 16 show that PCA preprocessing retains series principal information on Electricity and Traffic datasets, matching TSF performance with original series, and enabling training/inference acceleration.

Table 16: TSF experiments on the Electricity and Traffic datasets. The * symbols after models indicate the application of PCA before inputting the series into the models. Bold font represents the superior result.

| Models | | Linear | | Linear* | | Informer | | Informer* | | FEDformer | | FEDformer* | |
|---|---|---|---|---|---|---|---|---|---|---|---|---|---|
| Metric | | MSE | MAE | MSE | MAE | MSE | MAE | MSE | MAE | MSE | MAE | MSE | MAE |
| Electricity | 96 | 0.213 | 0.326 | **0.212** | **0.325** | **0.307** | **0.391** | 0.322 | 0.413 | 0.495 | 0.526 | **0.286** | **0.388** |
| | 192 | 0.241 | 0.347 | **0.240** | **0.344** | 0.341 | 0.420 | **0.347** | **0.426** | 0.434 | 0.492 | **0.314** | **0.404** |
| | 336 | 0.275 | 0.372 | **0.273** | **0.369** | 0.475 | 0.515 | **0.422** | **0.476** | 0.545 | 0.548 | **0.346** | **0.433** |
| | 720 | 0.312 | 0.414 | **0.306** | **0.409** | 0.644 | 0.611 | **0.537** | **0.539** | 0.566 | 0.572 | **0.463** | **0.504** |
| Traffic | 96 | **0.138** | **0.229** | 0.144 | 0.237 | 0.210 | 0.300 | **0.183** | **0.271** | 0.265 | 0.367 | **0.186** | **0.285** |
| | 192 | **0.141** | **0.231** | 0.146 | 0.238 | 0.221 | 0.325 | **0.189** | **0.280** | 0.270 | 0.371 | **0.191** | **0.288** |
| | 336 | **0.142** | **0.236** | 0.147 | 0.244 | 0.234 | 0.350 | **0.203** | **0.305** | 0.288 | 0.387 | **0.219** | **0.311** |
| | 720 | **0.156** | **0.251** | 0.167 | 0.265 | 0.305 | 0.420 | **0.253** | **0.328** | 0.305 | 0.408 | **0.230** | **0.336** |
| Better Count | | 8 | | 8 | | 2 | | 14 | | 0 | | 16 | |

## L  PCA VISUALIZATIONS

Fig. 7 depicts the shapes of series after PCA preprocessing and the series obtained by inverse transforming PCA series. It is evident that PCA series include the primary information of the original series with a small subset of initial values (principal components), while the remaining values exhibit minimal fluctuations. The similarity of the original series can also be reflected in the PCA series. Furthermore, series inverse transformed from PCA series appear significantly smoother compared to the original series, effectively achieving denoising of the series.

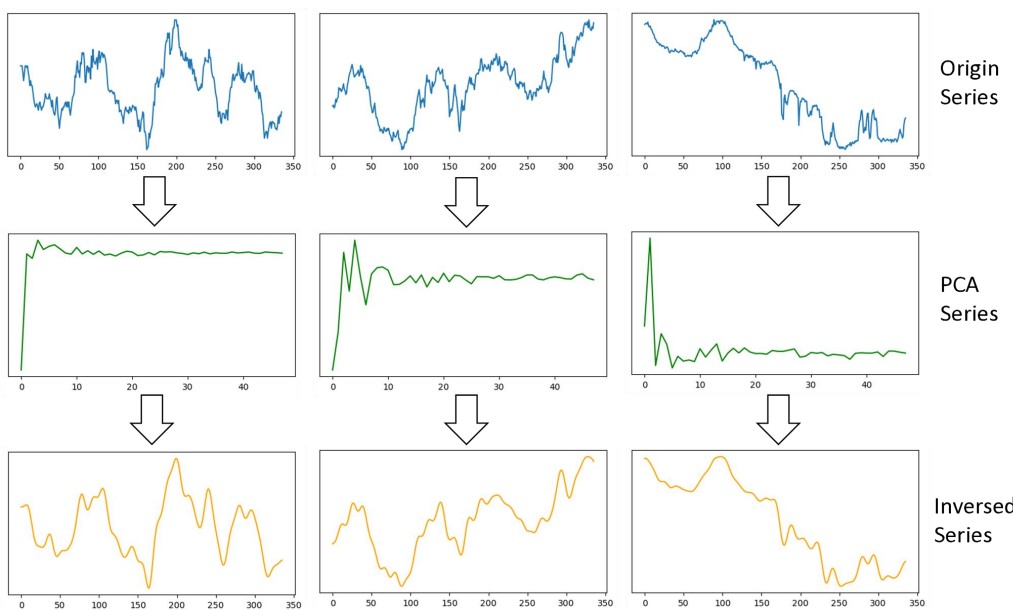

Figure 7: Visualizations of original series, PCA series and PCA-inversed series.

## M  PREDICTION SHOWCASES

Fig. 8 presents some prediction showcases of the Linear model with and without PCA preprocessing. It is observed that the predictions of the Linear model on the original series and the PCA series are highly consistent.

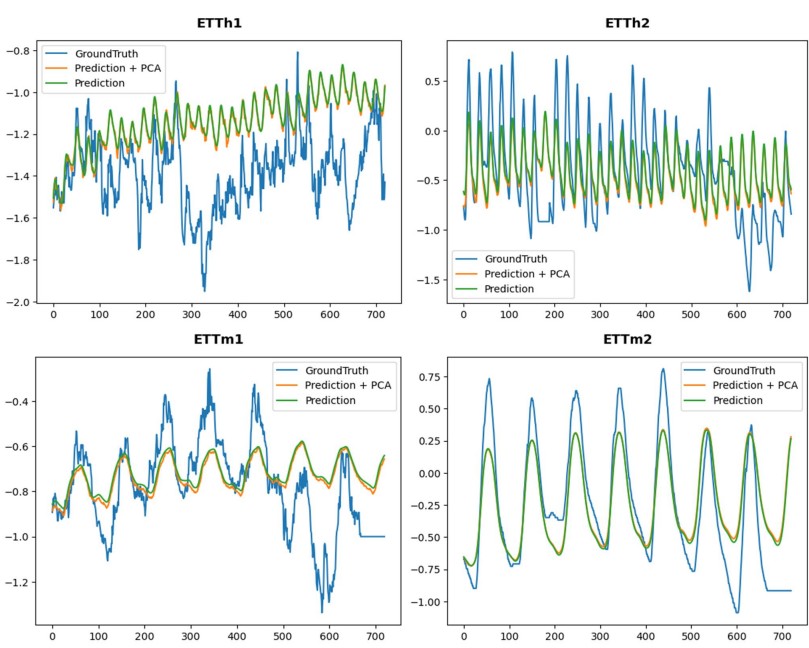

Figure 8: Prediction showcases on ETT datasets.

