# OpenReview forum: "Revisiting PCA for Time Series Reduction in Temporal Dimension"
_ICLR.cc/2025/Conference — Submitted to ICLR 2025_

### Official Review · Reviewer_ohhk · 2024-10-26

**Soundness:** 3
**Presentation:** 3
**Contribution:** 2
**Rating:** 5
**Confidence:** 2

**Summary:**

The paper investigates using Principal Component Analysis (PCA) to reduce the temporal dimensionality of time series data in deep learning models for tasks like classification, forecasting, and regression. Traditionally, PCA has been applied to reduce variable dimensions, but this study applies PCA across time windows, aiming to maintain temporal structure while reducing redundancy and computational costs. Results show that PCA preprocessing can accelerate training and inference by up to 40% in some models, like Informer, and reduce memory usage by 30% in models like TimesNet without compromising accuracy. PCA also proves effective in noise reduction, retaining essential statistical characteristics and supporting efficient learning across different deep learning models.

**Strengths:**

- It clearly introduces the problem of temporal dimensionality reduction in time series data and provides a solid rationale for using PCA.
- The experimental setup is thorough, covering various time series tasks (classification, forecasting, and regression) and a range of model types (Linear, Transformer, CNN, RNN), which effectively illustrates the generalizability of the approach.
- The paper strengthens its argument by presenting concrete metrics, such as GPU memory reduction and speed improvements.

**Weaknesses:**

- The intuition behind why PCA is specifically suitable for time series dimensionality reduction is not clearly explained. Many studies have shown that using orthogonal bases (e.g., FFT, wavelets, Legendre polynomials) can improve performance and reduce dimensionality, yet the paper does not address how PCA differs or why these methods were not included in comparisons.
- Adding comparisons with modern compression techniques, beyond linear methods and downsampling, could make the evaluation more robust.
- Some sections, particularly on the theoretical underpinnings of PCA’s use for time series, could benefit from clearer explanations to aid reader comprehension.
- Each table could benefit from explanations of the metrics used, clarifying what constitutes a “good” or “bad” result (e.g., lower MSE is better), which would help readers interpret the results more easily.
- More detailed visualizations, such as diagrams showing PCA’s effects on time series structure and feature retention, could enhance clarity.

**Questions:**

What makes PCA particularly effective here? Is there something unique about the space spanned by its vectors?

How does the dimensionality affect the results? A graph showing MSE versus the number of dimensions (n) would be helpful.

When selecting the first n eigenvalues, do you choose the largest, the smallest, or select them randomly?

---

> ### Author Response · Authors · 2024-11-22
>
> We deeply appreciate the insightful review and will do our utmost to address the questions and weaknesses.
>
> W1&W3: Theoretical underpinnings of PCA.
> Thank you for your comments. When we initially discovered that PCA could effectively reduce the temporal dimensionality of time series data, we found it intriguing. We searched for theoretical support in existing literature but did not find any relevant references, possibly due to the counterintuitive nature of our findings. Therefore, we explained this phenomenon through a combination of PCA mechanism analysis and visualization techniques. Specifically, PCA can effectively denoise and retain the primary statistical characteristics of the data. This is supported by the principles of PCA, its applications in other domains, and the visualizations we provide. Regarding why PCA can maintain model performance despite disrupting trends and periodicities, we speculate that the presence of specific trends or periodicities in historical series is not necessarily essential for the learning process of TSA models, Instead, the presence of consistent and coherent patterns is sufficient for models to provide accurate predictions. We offer the following detailed explanation: If we assume that all historical windows in the training set exhibit an increasing trend, and we simultaneously change them to a decreasing trend while keeping the trend of the target series unchanged (also assumed to be an increasing trend), this would not significantly affect the model's learning. Essentially, the model would learn that a decreasing trend in historical series can lead to an increasing trend in future series, rather than an increasing trend leading to an increasing trend. Similarly, applying the same transformation or scaling to the periodic information in all historical windows in the training set would not significantly impact the model's learning. Therefore, although PCA may alter the trend or periodicity, it introduces new coherent patterns—such as the main directions of variation, denoised low-dimensional representations, and latent features. These new consistent features in the training set enable the model to learn effectively.
>
> W1&W2: Comparation with other frequency-based dimensional reduction techniques.
> A2: Thanks you for your useful suggestions, and we have compared PCA with FFT and DWT as your suggestion. In the experiments, the original series is first transformed from the time domain to the frequency domain using either FFT or DWT. The top k frequency components (where k is 48, the same as the number of principle components) are then selected and input these into the TSA models. The results are shown in Table A. It is evident that the top k frequency components obtained using FFT or DWT fail to accurately capture the key information in the original series and effectively compress the series, leading to a significant decrease in model performance. We have also included these contents in Section G of the Supplemental Materials.
>
> Table A: Comparison of PCA with FFT and DWT as series reduction methods. Lower MSE/MAE indicates better performance. Bold font represents the superior result.
> |Method|||Linear||PCA||FFT||DWT|
> |-|-|-|-|-|-|-|-|-|-|
> |Dataset|Length|MSE|MAE|MSE|MAE|MSE|MAE|MSE|MAE|
> |ETTm1|96|**0.028**|**0.125**|0.029|0.126|2.110|1.328|1.827|1.299|
> ||192|0.043|0.154|**0.042**|**0.151**|2.086|1.318|1.943|1.344|
> ||336|0.059|0.180|**0.056**|**0.176**|2.205|1.356|1.767|1.279|
> ||720|**0.080**|**0.211**|0.081|0.212|2.232|1.348|1.981|1.358|
> |ETTm2|96|0.066|0.189|**0.065**|**0.188**|3.417|1.467|1.330|1.010|
> ||192|0.094|0.230|**0.092**|**0.228**|3.883|1.566|1.460|1.068|
> ||336|**0.120**|**0.263**|0.123|0.267|3.273|1.442|1.421|1.049|
> ||720|0.175|**0.320**|**0.174**|**0.320**|3.371|1.465|1.572|1.111|
> |Better Count|||7||10||0||0|
>
> W4: Explanations of the metrics.
> A4: Thank you for your valuable feedback. We appreciate your suggestion to enhance the clarity of our tables by providing explanations of the metrics used and clarifying what constitutes a "good" or "bad" result. For TSF and TSER tasks, the metrics MSE, MAE, and RMSE are all better when lower. For TSC tasks, the metric Accuracy is better when higher. In the revised paper, we have indicated this in the captions of tables.
>
> W5: Detailed visualizations.
> A5: Thanks for your comments. The detailed effects of PCA are illustrated in Figure 7 of the Supplemental Materials. We previously misunderstood and thought that the Supplementary Material needed to be submitted separately from the main text, so we placed it in a zip file. In the updated version, we have integrated the Supplementary Material with the main text. From Figure 7 we can see that PCA series include the primary information of the original series with a small subset of initial values (principal components), while the remaining values exhibit minimal fluctuations.

---

> ### Author Response · Authors · 2024-11-22
>
> Q1: PCA’s effectiveness.
> A6: Thank you for your valuable review. PCA transforms the original series into a different space by projecting them onto a new set of axes defined by the principal components. It retains only the most significant principal components while discarding the less important ones, which serves as a noise filtering mechanism. Additionally, since PCA only performs a spatial transformation, many of the statistical characteristics of the original series, such as mean, peak values, and higher-order moments, are preserved. This ensures that the transformed data retains key properties of the original data, which can be crucial for time series analysis. Moreover, while PCA can filter noise and retain statistical information, it also disrupts the periodicity and trends of the original series. However, we discuss from another perspective that periodicity and trends are not necessarily essential for time series analysis. Instead, the presence of consistent and coherent patterns is sufficient for models to provide accurate predictions.
>
> Q2: Impact of the number of dimensions on model’s performance.
> A7: Thanks for your comments. The Impact of the number of dimensions on model’s performance is illustrated in Figure 6 of the Supplemental Materials. In the updated version, we have integrated the Supplementary Material with the main text. From Figure 6 we can see that as the number of principal components increases, the importance of the selected features also increases, but the rate of increase diminishes. However, after the number of principal components reaching to 48 (the number chosen in our experiment), further increasing the number of principal components results in minimal change in feature importance.
>
> Q3: Eigenvectors.
> A8: When selecting eigenvectors, we choose the largest n eigenvectors because they correspond to the directions of maximum variance in the data. By retaining these eigenvectors, we ensure that the transformed data captures the most significant features and information, thereby effectively reducing the dimensionality while minimizing information loss.

---

> ### Author Response · Authors · 2024-11-28
>
> Dear Reviewer ohhk,
>
> We would appreciate it if you could let us know whether our responses have adequately addressed your concerns. We are happy to address any further questions or concerns you may have. Thank you!
>
> Best wishes,
> Paper6918 Authors

---

> ### Comment · Area_Chair_GndG · 2024-12-02
> **Please provide a response to the authors of submission 6918**
>
> Dear Reviewer ohhk,
>
> The above comment to Reviewer KQnv also applies to you, as the authors have provided an extensive response to your review, including additional experimental results.
>
> The discussion period is almost over, so please read the responses the authors of submission 6918 have provided to your review.
> Please specify which of your concerns were addressed and explain your decision to update or not update your score.
>
> All the best,
>
> The AC

---

### Official Review · Reviewer_KQnv · 2024-10-29

**Soundness:** 2
**Presentation:** 2
**Contribution:** 2
**Rating:** 3
**Confidence:** 5

**Summary:**

The authors propose an innovative approach for preprocessing time series data by applying Principal Component Analysis (PCA) to data within sliding windows. This method aims to extract the principal components from the data, effectively reducing dimensionality before feeding it into a deep learning network. Traditionally, it is commonly believed that applying PCA along the time dimension can disrupt temporal dependencies inherent in time series data. Contradicting this notion, the authors suggest that applying PCA to sliding sequence windows can preserve model performance while enhancing computational efficiency. They support their claims with experiments conducted on three primary tasks: time series classification, prediction, and regression.

**Strengths:**

The idea of applying PCA within sliding windows offers a fresh perspective on dimensionality reduction for time series data. By reducing the dimensionality of the input data, the proposed method can decrease computational load, which is particularly beneficial for deep learning models dealing with large-scale or high-frequency time series data.

**Weaknesses:**

The study exhibits several weaknesses. Firstly, it lacks clarity on dimensionality reduction along the time dimension, focusing instead on feature dimension reduction without reducing time steps, which contradicts its stated goal. Secondly, the application of PCA to the test data in classification tasks is ambiguous; applying PCA to test data is inappropriate, but without it, the claimed acceleration in inference time lacks justification. Thirdly, the study misrepresents related work by critiquing standard practices designed to prevent information leakage, and its theoretical analysis fails to support the core claim of time dimension reduction. Additionally, the experiments lack statistical validation and parameter exploration, relying on single runs with fixed principal component numbers, raising concerns about generalizability and potential overfitting. The authors make overgeneralized and absolute claims about the benefits of PCA without sufficient evidence, ignoring observed performance degradation in certain datasets. Furthermore, limited dataset diversity suggests results may be dataset-specific, and discrepancies in reported data raise doubts about reliability. Lastly, the study challenges established concepts in time series analysis without adequate empirical support, and methodological inconsistencies, such as varying the number of principal components without clear rationale, hinder reproducibility and limit the applicability of the findings.

**Questions:**

1. The paper does not seem to demonstrate whether the number of time steps is reduced or to what extent. It only generally mentions at the beginning that applying PCA will compress the time series, but it is unclear whether the compression target is the time steps or the data component features within the sliding window. In the Introduction, the authors state that dimensionality reduction techniques for time series data mainly focus on the variable dimension, and they intend to apply PCA for dimensionality reduction along the time dimension. However, from the overall description, the authors appear to only apply PCA to the time-step data within the sliding window to extract local features. This method extracts feature information from each window, but the number of time steps within the window seems to remain unchanged.

2. It is currently unclear whether the authors also applied PCA to the test set in the classification task. If the authors used PCA to preprocess the test set, this would be unreasonable because the test data should be assumed to be unknown beforehand. If the authors did not apply PCA to the test set, maintaining the original data format and attributes while keeping the network unchanged, then theoretically, there should not be a significant acceleration in inference time.

3. There is an issue with unreasonable descriptions in the related work section. The authors discuss the limitations of Xu et al.'s work titled "Transformer multivariate forecasting: Less is more?" In their second point, they state: "Secondly, it is designed for scenarios where a multivariate series forecasts a univariate series, focusing on reducing the variable dimension of covariate series without preprocessing the target variable series, even if the covariate series may have minimal association with the target series."

4. In the theoretical analysis section, as shown in Figure 3, the authors only demonstrate the effectiveness of PCA in reducing dimensionality along the feature dimension. However, they do not address dimensionality reduction along the time dimension (i.e., the compression of the number of time steps).

5. It appears that the experimental results presented in Figure 3 are based on a single experiment. The authors did not verify the generalizability of their results by experimenting with different numbers of principal components, k. Without varying k, there's a risk that the chosen value might be a "lucky number" that coincidentally yields favorable results.

6. The experimental results may be influenced by the characteristics of the specific dataset used. The smoothing effect observed in Figure 3 might only be applicable to the current dataset and may not represent the performance on other time series data. Including experiments on a variety of datasets could improve the credibility of the conclusions.

7.The authors propose that "Specific trends and periodic patterns in historical series may not be crucial for the learning of TSA models." In the field of Time Series Analysis (TSA), traditional viewpoints and a large body of research emphasize the importance of trends and periodic patterns. These elements are critical for understanding the inherent structure of the data and for predicting future values.

8. The authors' statement, "Therefore, although PCA may alter the trend or periodicity, it introduces new coherent patterns—such as the main directions of variation, denoised low-dimensional representations, and latent features—that benefit TSA model learning without negatively impacting predictive performance," is overly absolute. Claiming that there are no negative impacts is too definitive. In practical applications, any data transformation can potentially have both positive and negative effects on model performance; the specific outcome depends on the characteristics of the data and the type of model employed.

9.  It seems that the experiments presented in Tables 2, 3, and 4 are based on single runs without any statistical significance testing. There is no indication of whether the results are consistent across multiple trials or if they could be due to random chance. Furthermore, in each experiment, the number of principal components (k) selected for PCA is based on a single value, and this value differs across different datasets.

10.  In Table 2, concerning the Time Series Classification (TSC) experiments, the authors conclude based on the results: "These results reveal PCA’s efficacy in extracting series information for TSC tasks without performance loss, enabling faster training/inference." However, the results presented in Table 2 indicate that applying PCA on certain datasets and networks can lead to significant performance degradation. For example, on the SelfRegulationSCP1 dataset, the accuracy of the TimesNet network decreased by 23.2% after applying PCA. This substantial drop contradicts the authors' absolute assertion of "without performance loss." Out of the 20 metrics reported, only 10 show performance improvement when PCA is applied, which amounts to just 50%. This proportion raises doubts about the claim made in the abstract that applying PCA to sliding sequence windows can maintain model performance.

11.The authors state in Table 3: "The results of Linear are adapted from the study (Zeng et al., 2023)." However, upon reviewing the cited paper by Zeng et al. (2023), I was unable to locate the specific data presented by the authors. This discrepancy raises concerns about the reliability and accuracy of the data used in their experiments.

---

> ### Author Response · Authors · 2024-11-22
>
> We deeply appreciate the insightful review and will do our best to address the questions and weaknesses.
>
> W1&Q1: Dimensionality reduction along the time dimension.
> A1: We apologize for any confusion. We would like to clarify that throughout the paper, PCA is specifically applied for dimensionality reduction along the time dimension in time series data. All our experiments involve using PCA to reduce the dimensionality of the original time series (compressing the time steps) before feeding the reduced series into the models. For classification tasks, the original series lengths (time steps) range from 152 to 1751, which we reduce to 16, 48, or 96. For forecasting tasks, the original series length is 336, which we reduce to 48. For extrinsic regression tasks, the original series lengths are 84 or 266, which we reduce to 16 and 48, respectively.
>
> W2&Q2: PCA in classification tasks.
> A2: We apologize for any inconvenience in understanding. For the three TSA tasks, PCA is initially fitted on the training data. During inference, the pre-fitted PCA model is applied to the test data without re-fitting, resulting in a significant reduction in inference time, as demonstrated in Figure 4 and Table 9.
>
> W3&Q3: Description of the related work.
> A3: Thank you for your comments. We have re-read the paper carefully, but we did not find errors in our understanding or description of the paper. First, the experiments in the paper use multiple variables to predict a single target variable. Second, the method in that paper does not process the target variable series; instead, it reduces the dimensionality of the M covariate series to P series, without shortening the length of each series. Third, we merely speculate that other covariates series may have limited utility for predicting the target variable, and there are many studies [1,2] show that channel-independent is more effective for time series forecasting. However, we did not claim that the covariate series are definitely useless for predicting the target series. We would greatly appreciate it if you could provide detailed feedback on any inaccuracies in our descriptions and offer specific reasons.
>
> [1] Zeng, Ailing, et al. "Are Transformers Effective for Time Series Forecasting?." arxiv preprint arxiv:2205.13504 (2022).
> [2] Nie, Yuqi, et al. "A time series is worth 64 words: Long-term forecasting with transformers." arxiv preprint arxiv:2211.14730 (2022).
>
> Q4: Figure 3 and Dimensionality reduction along the time dimension.
> A4: We apologize for any confusion. We would like to respectfully clarify that in our paper, PCA is applied to the time dimension rather than the feature dimension. In our study, each time step is considered a 'feature' for PCA process.  Additionally, Figure 3(a) shows the effect of applying PCA to the series and then inversely transforming them back to the original series. The PCA-inversed series is significantly smoother than the original series, indicating that PCA effectively filters out noise while preserving essential features. Figure 3(b) is intended to demonstrate that the distribution of statistical characteristics of the PCA-inversed series are similar to those of the original series. Neither of these subplots aims to demonstrate the effectiveness of PCA in reducing dimensionality along the feature dimension or the time dimension.
>
> Q5: Figure 3 and impact of the number of dimensions on model’s performance.
> A5: Thanks for your comments. Figure 3 is a schematic diagram intended to illustrate the denoising effect of PCA and its ability to retain the statistical information of the original series, rather than to present experimental results. The Impact of the number of dimensions k on model’s performance is illustrated in Figure 6 of the Supplemental Materials. In the updated version, we have integrated the Supplementary Material with the main text. From Figure 6 we can see that as the number of principal components k increases, the importance of the selected features also increases, but the rate of increase diminishes. However, after k reaching to 48 (the number chosen in our experiment), further increasing k results in minimal change in feature importance.

---

> ### Author Response · Authors · 2024-11-22
>
> Q6: Figure 3 and more datasets.
> A13: Thanks for your suggestions. We conducted tests on more datasets and found that PCA has similar effects in denoising and retaining the statistical information of the original series. Specifically, we have added the experiments on multiple UCR datasets for TSC tasks. The results in Table A show that PCA preprocessing retains the principal information of the series on the UCR datasets, matches the TSC performance of the original series, and enables faster training/inference. And we also applied PCA to the commonly used TSF datasets, Electricity and Traffic. The results in Table B show that PCA preprocessing retains series principal information on Electricity and Traffic datasets, matching TSF performance with original series, and enabling training/inference acceleration.
>
> Table A: TSC experiments on UCR datasets. The * symbols after models indicate the application of PCA before inputting the series into the models. Accuracy metric is adopted. Bold font represents the superior result.
> |Dataset|Linear|Linear*|Informer|Informer*|FEDformer|FEDformer*|
> |-|-|-|-|-|-|-|
> |ACSF1|0.400|**0.580**|0.640|**0.780**|0.560|**0.730**|
> |Adiac|0.684|**0.760**|0.538|**0.716**|0.560|**0.729**|
> |ChlorineConcentration|0.553|**0.771**|0.564|**0.722**|**0.607**|0.544|
> |Computers|0.536|**0.600**|0.628|**0.640**|**0.830**|0.648|
> |Earthquakes|0.597|**0.691**|**0.748**|0.719|0.734|**0.755**|
> |ElectricDevices|**0.482**|0.479|**0.695**|0.605|**0.645**|0.563|
> |GunPointAgeSpan|0.864|**0.892**|0.889|**0.930**|0.775|**0.892**|
> |GunPointMaleVersusFemale|0.731|**0.991**|**0.997**|**0.997**|0.706|**0.991**|
> |GestureMidAirD1|0.477|**0.500**|0.431|**0.515**|**0.692**|0.500|
> |GestureMidAirD2|**0.485**|0.454|**0.523**|0.400|0.346|**0.415**|
> |GestureMidAirD3|**0.323**|0.254|**0.377**|0.277|0.231|**0.292**|
> |AllGestureWiimoteX|**0.296**|0.283|0.289|**0.403**|**0.460**|0.384|
> |AllGestureWiimoteY|0.319|**0.324**|**0.516**|0.387|0.409|**0.424**|
> |AllGestureWiimoteZ|**0.320**|**0.320**|0.296|**0.372**|**0.480**|0.366|
> |FordA|0.504|**0.507**|0.523|**0.817**|0.639|**0.822**|
> |FordB|0.532|**0.546**|0.549|**0.709**|0.672|**0.685**|
> |Better Count|5|12|6|11|6|10|
>
> Table B: TSF experiments on the Electricity and Traffic datasets. Bold font represents the superior result.
> Method||Linear||Linear*||Informer||Informer*||FEDformer||FEDformer*||
> |-|-|-|-|-|-|-|-|-|-|-|-|-|-|
> Dataset|Length|MSE|MAE|MSE|MAE|MSE|MAE|MSE|MAE|MSE|MAE|MSE|MAE|
> |Electricity|96|0.213|0.326|**0.212**|**0.325**|**0.307**|**0.391**|0.322|0.413|0.495|0.526|**0.286**|**0.388**|
> ||192|0.241|0.347|**0.240**|**0.344**|0.341|0.420|**0.347**|**0.426**|0.434|0.492|**0.314**|**0.404**|
> ||336|0.275|0.372|**0.273**|**0.369**|0.475|0.515|**0.422**|**0.476**|0.545|0.548|**0.346**|**0.433**|
> ||720|0.312|0.414|**0.306**|**0.409**|0.644|0.611|**0.537**|**0.539**|0.566|0.572|**0.463**|**0.504**|
> |Traffic|96|**0.138**|**0.229**|0.144|0.237|0.210|0.300|**0.183**|**0.271**|0.265|0.367|**0.186**|**0.285**|
> ||192|**0.141**|**0.231**|0.146|0.238|0.221|0.325|**0.189**|**0.280**|0.270|0.371|**0.191**|**0.288**|
> ||336|**0.142**|**0.236**|0.147|0.244|0.234|0.350|**0.203**|**0.305**|0.288|0.387|**0.219**|**0.311**|
> ||720|**0.156**|**0.251**|0.167|0.265|0.305|0.420|**0.253**|**0.328**|0.305|0.408|**0.230**|**0.336**|
> |Better Count|||8||8||2||14||0||16|
>
> Q7: Explanation of trends and periodic patterns in historical series.
> A7: Thanks for your comments. Applying PCA to time series disrupts the original periodicity and trends. However, through experiments, we found that despite this disruption, the model still achieves similar performance as before. We have provided our explanations of this phenomenon: if we assume all historical windows in the training set exhibit an increasing trend, and we simultaneously change them to a decreasing trend, while keeping the trend of the target series unchanged (also assumed to be an increasing trend), this would not affect the model's learning. Essentially, the model would learn that a decreasing trend in historical series can lead to an increasing trend in future series, rather than an increasing trend leading to an increasing trend. Similarly, applying the same transformation or scaling to the periodic information in all historical windows in the training set would not greatly impact the model's learning. Through the observations, we want to show that the presence of specific trends/periodicities in historical series is not necessary for the learning process of TSA models. Instead, the presence of consistent and coherent patterns is sufficient for models to provide accurate predictions. Therefore, although PCA may alter the trend or periodicity, it introduces new coherent patterns (equivalent to applying the same transformations to all historical windows)—such as the main directions of variation, denoised low-dimensional representations, and latent features. These new consistent features in the training set enable the model to learn effectively.

---

> > ### Author Response · Authors · 2024-11-22
> >
> > Q8: Expression issue.
> > A8: Thank you for pointing out the issue here. You are correct that our original expression was absolute. We have revised the expression as follows: Therefore, although PCA may alter the trend or periodicity, it introduces new coherent patterns—such as the main directions of variation, denoised low-dimensional representations, and latent features—that effectively benefit TSA model learning.
> >
> > Q9: 5-run average.
> > A9: We apologize for not clarifying this in the paper. All our results are based on a 5-run average, which is consistent with other top-tier works (e.g., iTransformer, MR-Diff, DLinear).
> >
> > Q10: Classification performance in Table 2.
> > A10: Thank you for your careful observation. Despite the accuracy of the TimesNet network decreasing by 23.2% after applying PCA, the accuracy of the FEDformer network increased by 25.0% after applying PCA on the SelfRegulationSCP1 dataset. We speculate that the instability in PCA application on the SelfRegulationSCP1 dataset can be attributed to two main reasons: First, the SelfRegulationSCP1 dataset has fewer samples, which introduces some randomness in the results. Additionally, different models have varying difficulties in capturing the features of different datasets. For a specific model, PCA preprocessing may make some datasets easier to learn while making others more difficult. However, from the overall results of classification, forecasting, and regression, PCA preprocessing does not degrade model performance and can accelerate training and inference while reducing memory usage.
> >
> > Q11: Results in Table 3.
> > A11: The results for the linear model in Table 3 are taken from Table 9 of study [1] (arXiv version). Additionally, in Table 9 of study [1], the historical window length for Informer and FEDformer is 96, which does not fully leverage their performance capabilities. We retested these models with a historical window length of 336 to better utilize their potential.
> >
> > [1] Zeng, Ailing, et al. "Are Transformers Effective for Time Series Forecasting?." arxiv preprint arxiv:2205.13504 (2022).

---

> > > ### Author Response · Authors · 2024-11-28
> > >
> > > Dear Reviewer KQnv,
> > >
> > > We would appreciate it if you could let us know whether our responses have adequately addressed your concerns. We are happy to address any further questions or concerns you may have. Thank you!
> > >
> > > Best wishes,
> > > Paper6918 Authors

---

> ### Comment · Area_Chair_GndG · 2024-12-02
> **Please provide a response to the authors of submission 6918**
>
> Dear Reviewer KQnv,
>
> The discussion period is almost over, so please read the responses the authors of submission 6918 have provided to your review.
>
> Please specify which of your concerns were addressed and explain your decision to update or not update your score.
>
> All the best,
>
> The AC

---

### Official Review · Reviewer_bfos · 2024-11-01

**Soundness:** 1
**Presentation:** 3
**Contribution:** 2
**Rating:** 5
**Confidence:** 4

**Summary:**

This paper explores the application of Principal Component Analysis (PCA) for dimensionality reduction of the temporal dimension. The authors argue that PCA's ability to reduce dimensionality enables the extraction of essential features underlying time series, thereby improving the efficiency of downstream tasks. Experimentally, the study applies forecasting, classification, and extrinsic regression tasks to the PCA-learned representations. The results show significant improvements in computational time and memory consumption compared to purely supervised approaches applied directly to the raw time series.

**Strengths:**

- S1. I think it is interesting to use unsupervised learning as a first step to reduce memory and computation time for downstream supervised tasks. This approach could be particularly beneficial when dealing with large amounts of time series data with numerous timestamps.

- S2. The paper is well written and easy to follow, making the concepts and methods presented clear and accessible to the reader.

- S3. The experimental results show that PCA accelerates both the training and inference processes while reducing the GPU memory usage for the considered downstream tasks.

**Weaknesses:**

- W1. A significant weakness of the paper is its lack of discussion and comparison with other representation learning methods.
     - Several claims in the paper appear to be inaccurate, such as: "To the best of our knowledge, there has been no systematic method for compressing time series data in the temporal dimension while preserving critical information" and "far less attention has been given to reducing the temporal dimension, despite the potential benefits of alleviating the burdens associated with processing long time series." In recent years, various unsupervised time series methods have effectively addressed this issue. For example, T-Loss [1] was one of the first models to fully compress the temporal dimension by leveraging contrastive learning and an encoder-only architecture. Another contrastive method, TS2Vec [2], learns representations that can be used for forecasting and classification in subsequent stages. Additionally, methods based on autoencoders with vector quantization [3,4] have demonstrated the ability to compress the temporal dimension by learning the core features of time series data.
    - The use of PCA representation does not appear to enhance the performance of the supervised model. While the authors argue that PCA representation accelerates training and inference (and reduces memory usage), the omission of other representation learning methods—such as a basic convolutional encoder-decoder—makes it difficult to fully evaluate the contribution of this paper.


- W2. From an experimental perspective, several aspects seem questionable.
    - For the classification tasks, the authors selected a few datasets from the UEA and applied PCA pairs with models that are primarily known as forecasting baselines (except for TimesNet). The reported results, whether with or without PCA, do not represent state-of-the-art performance. It would have been beneficial to include models like InceptionTime or a simple ResNet for comparison.
    - In the forecasting tasks, the authors focused solely on the ETT datasets, which are recognized for their difficulty in forecasting. It would be more insightful to conduct similar experiments on datasets such as traffic or electricity, which may provide additional context and validation for the proposed methods.


[1] Unsupervised scalable representation learning for multivariate time series, Neurips 2019

[2] Ts2vec: Towards universal representation of time series, AAAI 2022

[3] Vector Quantized Time Series Generation with a Bidirectional Prior Model, AISTATS 2023

[4] Interpretable time series neural representation for classification purposes, IEEE DSAA 2023

**Questions:**

- Q1. Could you please include a comparative analysis section in their paper, directly comparing PCA's performance, computational efficiency, and memory usage against the methods you mentioned (T-Loss, TS2Vec, and autoencoder-based approaches). This would provide a clearer context for evaluating PCA's contribution relative to recent advances in the field.

- Q2. Coud you please include state-of-the-art classification models like InceptionTime and ResNet in their comparison for the classification tasks, providing a stronger baseline for evaluating PCA's impact.

- Q3. I Believe that it would be valuable to expand the forecasting experiments to include additional widely-used datasets such as traffic and electricity, alongside the ETT datasets. Suggest specific datasets that are commonly used in the field for benchmarking.

---

> ### Author Response · Authors · 2024-11-22
>
> We sincerely appreciate the valuable review and will do our best to address the questions and weaknesses.
>
> W1&Q1: Comparation with other representation learning methods.
> A1: Thank you for your insightful comments and references. We have carefully read the methods you mentioned. While these representation learning methods are effective for feature extraction and series compression in specific scenarios, they are not general-purpose approaches and cannot be easily integrated with arbitrary time series models or tasks, and obtaining time series representations (compressing time series) is just the initial part of these methods. For example, in the TS2Vec framework, an SVM classifier is required for TSC tasks, while a ridge regression model is needed for TSF tasks. Therefore, these methods are more specialized models rather than pluggable, general-purpose frameworks or modules, which is inconsistent with our goal for PCA. Additionally, the primary objective of these representation learning methods is to learn better representations rather than to accelerate training/inference. As a result, they typically do not optimize for training efficiency or memory usage as extensively as PCA does. As your suggestions, we compared PCA with these representation learning-based methods on TSC tasks. As shown in Table A, the Linear + PCA model achieved the best performance in most settings. Additionally, PCA requires less training/inference overhead and less GPU memory compared to representation learning methods, as shown in Table B. We have also included these contents in Section I of the Supplemental Materials.
>
> Table A: Comparation of PCA with T-Loss, TS2Vec, and TimeVQVAE on TSC experiments. The accuracy metric is adopted. Bold font is the superior result.
> || Linear+PCA | T-Loss | TS2Vec | TimeVQVAE |
> | - | - | - | - | - |
> | EthanolConcentration | **0.300** | 0.289 | 0.287 | 0.203 |
> | Handwriting | 0.127 | 0.255 | **0.397** | 0.218 |
> | SelfRegulationSCP1 | **0.805** | 0.780 | 0.795 | 0.719 |
> | SelfRegulationSCP2 | **0.539** | 0.511 | 0.525 | 0.527 |
> | UWaveGestureLibrary | 0.409 | 0.622 | 0.666 | **0.668** |
> | Better Count | 3 | 0 | 1 | 1 |
>
> Table B: Computational efficiency, and memory usage comparation of PCA with T-Loss, TS2Vec, and TimeVQVAE. Bold font is the superior result.
> || Linear+PCA | T-Loss | TS2Vec | TimeVQVAE |
> | - | - | - | - | - |
> | Training time (s) | **14.82** | 302.50 | 25.92 | 62.98 |
> | Inference time (s) | **0.59** | 2.01 | 1.65 | 68.10 |
> | Memory usage (MiB) | **484** | 1290 | 2424 | 2870 |
>
> W2&Q2: Comparation with other classification baselines.
> A2: Thanks for your comments. In the top-tier study TimesNet[1], models such as Linear, FEDformer, and Informer are also used for TSC tasks, we also followed their experimental settings and tested the same models and datasets. As your suggestions, we applied PCA to SOTA classification models, InceptionTime[2] and ResNet[3]. The results are shown in Table C. It is evident that PCA is model-agnostic and remains effective even when applied to these SOTA classification models. We have also included these contents in Section J of the Supplemental Materials in the revised paper.
>
> Table C: TSC experiments of Inception and ResNet. The accuracy metric is adopted. The * symbols after models indicate the application of PCA before inputting the series into the models. Bold font is the superior result. PCA preprocessing retains series principal information, matching TSC performance with original series, and enabling training/inference acceleration.
> | Dataset | Inception | Inception* | ResNet | ResNet* |
> | - | - | - | - | - |
> | EthanolConcentration | 0.259 | **0.300** | 0.281 | **0.308** |
> | Handwriting | 0.075 | **0.119** | 0.076 | **0.105** |
> | SelfRegulationSCP1 | **0.833** | 0.758 | **0.867** | 0.754 |
> | SelfRegulationSCP2 | 0.489 | **0.561** | 0.528 | **0.539** |
> | UWaveGestureLibrary | **0.522** | 0.516 | **0.528** | 0.419 |
> | Better Count | 2 | 3 | 2 | 3 |
>
> [1] Wu, Haixu, et al. "Timesnet: Temporal 2d-variation modeling for general time series analysis." arxiv preprint arxiv:2210.02186 (2022).
> [2] https://github.com/TheMrGhostman/InceptionTime-Pytorch/blob/master/inception.py
> [3] https://github.com/hsd1503/resnet1d/blob/master/resnet1d.py

---

> > ### Author Response · Authors · 2024-11-22
> >
> > W2&Q3: TSF experiments on the Electricity and Traffic datasets.
> > A3: Thanks for your suggestions. As your suggestion, we applied PCA to the commonly used TSF datasets, Electricity and Traffic. The results in Table D show that PCA preprocessing retains series principal information on Electricity and Traffic datasets, matching TSF performance with original series, and enabling training/inference acceleration.
> >
> > Table D: TSF experiments on the Electricity and Traffic datasets. The * symbols after models indicate the application of PCA before inputting the series into the models. Bold font represents the superior result.
> >  Method||Linear||Linear* ||Informer||Informer* ||FEDformer||FEDformer* ||
> > |-|-|-|-|-|-|-|-|-|-|-|-|-|-|
> > |Dataset|Length|MSE|MAE|MSE|MAE|MSE|MAE|MSE|MAE|MSE|MAE|MSE|MAE|
> > |Electricity|96| 0.213 | 0.326 | **0.212** | **0.325** | **0.307** | **0.391** | 0.322 | 0.413 | 0.495 | 0.526 | **0.286** | **0.388** |
> > ||192| 0.241 | 0.347 | **0.240** | **0.344** | 0.341 | 0.420 | **0.347** | **0.426** | 0.434 | 0.492 | **0.314** | **0.404** |
> > ||336| 0.275 | 0.372 | **0.273** | **0.369** | 0.475 | 0.515 | **0.422** | **0.476** | 0.545 | 0.548 | **0.346** | **0.433** |
> > ||720| 0.312 | 0.414 | **0.306** | **0.409** | 0.644 | 0.611 | **0.537** | **0.539** | 0.566 | 0.572 | **0.463** | **0.504** |
> > |Traffic|96| **0.138** | **0.229** | 0.144 | 0.237 | 0.210 | 0.300 | **0.183** | **0.271** | 0.265 | 0.367 | **0.186** | **0.285** |
> > ||192| **0.141** | **0.231** | 0.146 | 0.238 | 0.221 | 0.325 | **0.189** | **0.280** | 0.270 | 0.371 | **0.191** | **0.288** |
> > ||336| **0.142** | **0.236** | 0.147 | 0.244 | 0.234 | 0.350 | **0.203** | **0.305** | 0.288 | 0.387 | **0.219** | **0.311** |
> > ||720| **0.156** | **0.251** | 0.167 | 0.265 | 0.305 | 0.420 | **0.253** | **0.328** | 0.305 | 0.408 | **0.230** | **0.336** |
> > |Better Count| | | 8 | | 8 | | 2 | | 14 | | 0 | | 16 |

---

> > > ### Comment · Reviewer_bfos · 2024-11-26
> > > **Comment**
> > >
> > > I appreciate the effort the authors have put into the rebuttal, and as a result I have increased my score from 3 to 5. However, I cannot recommend acceptance of the paper at this time. The paper remains insufficiently positioned within the broader literature on representation learning, and the additional discussion and experiments provided in the rebuttal are not sufficient to clearly establish a significant contribution.

---

> > > > ### Author Response · Authors · 2024-11-28
> > > >
> > > > Thank you for your response, further suggestions, and increasing our score. Our study aims to propose a general preprocessing method for time series, which we think has clear differences from representation learning-based approaches, and thus we did not extensively analyse related work in the field before. Based on your suggestion, we have added a comparison with the typical representation learning-based approaches in terms of performance and training/inference efficiency, which has further improved the quality of our paper. Moreover, our contributions extend beyond proposing a novel, general approach for dimensionality reduction along the time dimension for time series. We also challenge the traditional perception that PCA cannot be applied to data with temporal dependencies. This contribution broadens the applicability of the classic dimensionality reduction method PCA, enabling it to play a more significant role in a wider range of fields.
> > > >
> > > > Given these contributions, we kindly hope you will consider raising our score. We greatly appreciate your constructive feedback and have made every effort to integrate your suggestions to improve the clarity and quality of our work. Thank you for your time and thoughtful review.

---

### Official Review · Reviewer_EsNx · 2024-11-04

**Soundness:** 2
**Presentation:** 3
**Contribution:** 2
**Rating:** 5
**Confidence:** 4

**Summary:**

This manuscript revisits Principal Component Analysis (PCA) to explore its utility in reducing the temporal dimension of time series data, as a novel area of focus, because PCA has traditionally been applied mostly on the variable space. The paper posits that PCA, when applied to sliding series windows, not only maintains model performance but also enhances computational efficiency. Extensive experiments across time series classification, forecasting, and extrinsic regression tasks substantiate these claims. The paper suggests that PCA preprocessing results in reduced training and inference times without compromising on model effectiveness across various deep learning-based time series analysis models.

**Strengths:**

S1. This paper attempts to improve the efficiency of time series analysis tasks, which can be very useful for resource-constrained scenarios including edge computing.

S2. The evaluation was conducted on three different tasks, i.e., time series classification, forecasting, and extrinsic regression, demonstrating the versatility of the proposed methods.

S3. The paper is easy-to-parse.

**Weaknesses:**

W1. It is still hard to conclude from this paper that PCA before feeding into deep neural networks is a versatile solution that should be suggested to time series analysis tasks under resource constraints. Briefly speaking, neural networks, especially the early layers of neural networks, are considered as feature extraction as well as dimension adjusting. This overlaps a bit with the purpose of PCA.

W2. There are more dimensional reduction techniques for time series or high-dimensional vectors, rather than PCA itself. For example, DWT, FFT, etc. Although have been briefly discussed, it would still be very important to compare PCA with other deep model-agnostic dimension reduction techniques.

**Questions:**

Q1. The discussions provided in section 3.2 are not really theoretical analysis. The section title is a bit misleading. I would suggest to either break this section down to the motivation of the work, or rename it to some candidates like intuitional justification.

Q2. In line 340, why only 5 datasets from the UEA is selected, out of 30+ multivariate datasets? Also, the five datasets are finally precessed into univariate datasets, in which case why the original 100+ univariate datasets are excluded?

Q3. The accuracy reported in table 2 is pretty low on the first two datasets. And the differences with or without PCA are huge on some cases, e.g., FEDformer and TimesNet on SelfRegulationSCP1. Could the authors provide justification for these numbers? Otherwise, this would damage the versatility of the proposed approach.

Q4. What is the backbone model for the results in Table 5?

Q5. I am not sure if I fully understood line 302-304, “For example, if all positive trends…” Could the authors further explain a bit on this for me? Thanks!

---

> ### Author Response · Authors · 2024-11-22
>
> We deeply thank the valuable review and do our best to address questions and weaknesses.
>
> W1: The purpose of PCA.
> A1: Thanks for your comments. In our study, PCA is employed as a pluggable, general-purpose preprocessing technique for time series analysis (TSA), which can be integrated with various TSA models and applied to different downstream TSA tasks. While the early layers of neural networks may have similar effects, it's necessary to design different early layers for different neural networks to ensure that these layers can effectively extract series features and reduce dimensionality. Additionally, adding a dimensionality reduction layer at the beginning of the existing neural network might increase the training/inference burden and raise the risk of overfitting. Furthermore, we also experimented with adding a linear/1D-CNN dimension reduction layer before the original neural network to achieve dimensionality reduction. However, the experimental results shown in Table 6 indicate that their performance is inferior to that of PCA-based dimensionality reduction.
>
> W2: Comparation with other frequency-based dimensional reduction techniques.
> A2: Thanks you for your useful suggestions, and we have compared PCA with FFT and DWT as your suggestion. In the experiments, the original series is first transformed from the time domain to the frequency domain using either FFT or DWT. The top k frequency components (where k is 48, the same as the number of principle components) are then selected and input into the TSA models. The results are shown in Table A. It is evident that the top k frequency components obtained using FFT or DWT fail to accurately capture the key information in the original series and effectively compress the series, leading to a significant decrease in model performance. We have also included these contents in Section G of the Supplemental Materials.
>
> Table A: Comparison of PCA with FFT and DWT as series reduction methods. Bold font represents the superior result.
> |Method|||Linear||PCA||FFT||DWT|
> |-|-|-|-|-|-|-|-|-|-|
> |Dataset|Length|MSE|MAE|MSE|MAE|MSE|MAE|MSE|MAE|
> |ETTm1|96|**0.028**|**0.125**|0.029|0.126|2.110|1.328|1.827|1.299|
> ||192|0.043|0.154|**0.042**|**0.151**|2.086|1.318|1.943|1.344|
> ||336|0.059|0.180|**0.056**|**0.176**|2.205|1.356|1.767|1.279|
> ||720|**0.080**|**0.211**|0.081|0.212|2.232|1.348|1.981|1.358|
> |ETTm2|96|0.066|0.189|**0.065**|**0.188**|3.417|1.467|1.330|1.010|
> ||192|0.094|0.230|**0.092**|**0.228**|3.883|1.566|1.460|1.068|
> ||336|**0.120**|**0.263**|0.123|0.267|3.273|1.442|1.421|1.049|
> ||720|0.175|**0.320**|**0.174**|**0.320**|3.371|1.465|1.572|1.111|
> |Better Count|||7||10||0||0|

---

> ### Author Response · Authors · 2024-11-22
>
> Q1: Title of section 3.2.
> A3: Thanks for your helpful suggestion. Using "intuitional justification" as title would be more appropriate and we have updated the title in the revised paper.
>
> Q2: More TSC datasets.
> A4: Thanks for your comments. We adopted the TSC datasets used in the TimesNet[1] paper and selected series with long length for our experiments (series with short lengths have limited benefit from PCA dimensionality reduction). We have added the experiments on multiple UCR datasets based on your suggestions. The results in Table B show that the application of PCA doesn't decrease accuracy; Instead, it slightly improves performance in some cases. More importantly, the computational cost is greatly reduced, showing the effectiveness of PCA compression processing. This PCA processing aligns with the concept of Pareto optimization, improving or maintaining accuracy while greatly reducing computational resources. We have also included these contents in Section H of the Supplemental Materials.
>
> Table B: TSC experiments on the UCR datasets. The * symbols after models indicate the application of PCA before inputting the series into the models. The accuracy metric is adopted. Bold font represents the superior result.
> |Dataset|Linear|Linear*|Informer|Informer*|FEDformer|FEDformer*|
> |-|-|-|-|-|-|-|
> |ACSF1|0.400|**0.580**|0.640|**0.780**|0.560|**0.730**|
> |Adiac|0.684|**0.760**|0.538|**0.716**|0.560|**0.729**|
> |ChlorineConcentration|0.553|**0.771**|0.564|**0.722**|**0.607**|0.544|
> |Computers|0.536|**0.600**|0.628|**0.640**|**0.830**|0.648|
> |Earthquakes|0.597|**0.691**|**0.748**|0.719|0.734|**0.755**|
> |ElectricDevices|**0.482**|0.479|**0.695**|0.605|**0.645**|0.563|
> |GunPointAgeSpan|0.864|**0.892**|0.889|**0.930**|0.775|**0.892**|
> |GunPointMaleVersusFemale|0.731|**0.991**|**0.997**|**0.997**|0.706|**0.991**|
> |GestureMidAirD1|0.477|**0.500**|0.431|**0.515**|**0.692**|0.500|
> |GestureMidAirD2|**0.485**|0.454|**0.523**|0.400|0.346|**0.415**|
> |GestureMidAirD3|**0.323**|0.254|**0.377**|0.277|0.231|**0.292**|
> |AllGestureWiimoteX|**0.296**|0.283|0.289|**0.403**|**0.460**|0.384|
> |AllGestureWiimoteY|0.319|**0.324**|**0.516**|0.387|0.409|**0.424**|
> |AllGestureWiimoteZ|**0.320**|**0.320**|0.296|**0.372**|**0.480**|0.366|
> |FordA|0.504|**0.507**|0.523|**0.817**|0.639|**0.822**|
> |FordB|0.532|**0.546**|0.549|**0.709**|0.672|**0.685**|
> |BetterCount|5|12|6|11|6|10|
>
> [1] Wu, Haixu, et al. "Timesnet: Temporal 2d-variation modeling for general time series analysis." arXiv preprint arXiv:2210.02186 (2022).
>
> Q3: Accucary of Table 2.
> A5: Thanks for your careful observations.The first two datasets are challenging for TSC. We retested them and obtained the same results. We also verified the results reported in the TimesNet[1] and found that their method also struggled with classifying these datasets. We speculate that the following reasons contribute to this phenomenon: First, these datasets have fewer samples, which introduces some randomness in the results. Additionally, different models have varying difficulties in capturing the features of different datasets. For a specific model, PCA preprocessing may make some datasets easier to learn while making others more difficult. However, from the overall results of TSC, TSF, and TSER, PCA preprocessing does not degrade model performance and can accelerate training and inference while reducing memory usage.
>
> Q4: Backbone in Table 5.
> A6: We apologize for any inconvenience. The backbone model for the results in Table 5 is the Linear model.
>
> Q5: Explanation of positive trends.
> A7: We apologize for any inconvenience in understanding. Here is our detailed explanations: if we assume all historical windows in the training set exhibit an increasing trend, and we simultaneously change them to a decreasing trend, while keeping the trend of the target series unchanged (also assumed to be an increasing trend), this would not significantly affect the model's learning. Essentially, the model would learn that a decreasing trend in historical series can lead to an increasing trend in future series, rather than an increasing trend leading to an increasing trend. Similarly, applying the same transformation or scaling the periodic information in all historical windows in the training set would not significantly impact the model's learning.
> Through these observations, we want to show that the presence of specific trends or periodicities in historical series is not necessarily essential for the learning process of TSA models. Instead, the presence of consistent and coherent patterns is sufficient for models to provide accurate predictions. Therefore, although PCA may alter the trend or periodicity, it introduces new coherent patterns (equivalent to applying the same transformations to all historical windows)—such as the main directions of variation, denoised low-dimensional representations, and latent features. These new consistent features in the training set enable the model to learn effectively.

---

> ### Comment · Reviewer_EsNx · 2024-11-26
>
> Thanks to the authors for your detailed feedback with new empirical results. After very careful consideration, although I appreciate very much the new empirical evidence and agree that it has greatly improved the quality of the paper, I decided to maintain my original score.
>
> The major concern is that I am still not fully convinced by the main motivation, i.e., PCA should be taken as a versatile plug-in before feeding time series into neural networks. The main reason is that usually one would like to have all information fed into the neural network, which is usually carefully designed and will be carefully trained under proper constraints. Performing dimension reduction before feeding the input data into the neural network, is sort of implicitly suggesting that the neural network lacks the ability to identify the necessary information through the noise (which, at the same time, can be removed by the simple PCA method).
>
> I would suggest the authors to set up a stronger motivation for the adoption of PCA. One possible direction is considering the unique property of time series datasets, for example, the task-specific datasets are mostly pretty small but still with noise, while training neural networks is also notoriously hard.

---

> > ### Author Response · Authors · 2024-11-28
> >
> > Thank you for your response and further suggestions. We understand your concerns. Neural networks indeed possess a high generality; however, many preprocessing techniques can also be effectively combined with neural networks, including those you previously mentioned (DWT, FFT, etc). These preprocessing methods can reduce the learning difficulty for neural networks, leading to improved performance or higher training/inference efficiency. In the context of the TSA tasks we tested, PCA has proven to be an effective preprocessing tool for neural networks.
> >
> > Moreover, our contributions extend beyond proposing a novel, general approach for dimensionality reduction along the time dimension for time series. We also challenge the traditional perception that PCA cannot be applied to data with temporal dependencies. This contribution broadens the applicability of the classic dimensionality reduction method PCA, enabling it to play a more significant role in a wider range of fields.
> >
> > In light of these contributions, we respectfully hope that you could reconsider your evaluation. Once again, thank you for your response and valuable suggestions.

---

### Meta-Review · Area_Chair_GndG · 2024-12-21

**Metareview:**

The paper studies the application of PCA to data consisting of sliding windows from a time series, followed by the application of time series models to the dimensionality reduced data.

The reviewers appreciated that the approach is pertinent, that several tasks were considered and that the memory and computation time were reduced. However, serious concerns were raised about the claims of a theoretical analysis, the lack of comparison against other dimensionality reduction techniques for time series, the limited number of datasets considered and missing statistical significance.

While the authors have addressed some of the issues with additional experimental results on more datasets and comparisons with other methods, this was not sufficient to persuade the reviewers that the work is ready for publication, due to its insufficient positioning with respect to related work.

As I agree with the reviewers’ assessment, I do not recommend acceptance.

**Additional Comments On Reviewer Discussion:**

In the discussion, although the authors provided additional results, they ultimately failed to convincingly position their paper in the context of existing work. In the end, all 4 reviewers have opted to reject the paper, though only 2 of them participated in the discussion; the other 2 did not read the response even after being prompted to do so.

---

### Decision · Program_Chairs · 2025-01-22

Reject